# Carbon isotope budget indicates biological disequilibrium dominated ocean carbon storage at the Last Glacial Maximum

Anne Willem Omta[1] ✉, Christopher L. Follett[2,3], Jonathan M. Lauderdale [3] & Raffaele Ferrari [3]

Understanding the causes of the ~90 ppmv atmospheric $CO_2$ swings between glacial and interglacial climates is an important open challenge in paleoclimate research. Although the regularity of the glacial-interglacial cycles hints at a single driving mechanism, Earth System models require many independent physical and biological processes to explain the full observed $CO_2$ signal. Here we show that biologically sequestered carbon in the ocean can explain an atmospheric $CO_2$ change of $75 \pm 40$ ppmv, based on a mass balance calculation using published carbon isotopic measurements. An analysis of the carbon isotopic signatures of different water masses indicates similar regenerated carbon inventories at the Last Glacial Maximum and during the Holocene, requiring that the change in carbon storage was dominated by disequilibrium. We attribute the inferred change in carbon disequilibrium to expansion of sea-ice or change in the overturning circulation.

Measurements of atmospheric $CO_2$ trapped in ancient ice show that its concentration increased from roughly 190 ppmv at the Last Glacial Maximum (LGM; from 23,000 until 19,000 years before present) to ~280 ppmv during the pre-industrial Holocene[1]. Over the past 40 years, explanations for this deglacial $CO_2$ increase have relied primarily on numerical modeling studies. Early box models[2–4] suggested that rather modest changes in Southern Ocean nutrient utilization or air-sea gas exchange could account for the full 90 ppmv glacial-interglacial $CO_2$ change. However, studies with more comprehensive Earth System models found much weaker impacts of Southern Ocean processes on atmospheric $CO_2$[5,6]. The discrepancies have been attributed to the crude representation of the ocean circulation[7] and air-sea equilibration[8] in box models. The long-standing consensus is that while changes in Southern Ocean processes played an important role, several other concurrent physical, chemical and biological mechanisms are needed to explain the total ocean carbon uptake during glacial climates[9–13]. Furthermore, these mechanisms must be imposed externally one by one: models that allow $CO_2$ to evolve freely, without incorporating additional glacial mechanisms, often predict elevated atmospheric $CO_2$ concentrations at the LGM[14]. The requirement of many independent separate mechanisms is difficult to reconcile with the evidence that the magnitude of the glacial-interglacial $CO_2$ swings has been remarkably constant over the last few glacial cycles, appearing to demand that all the driving mechanisms be tightly coupled. Modeling the coupling between the various biogeochemical and physical mechanisms affecting the carbon cycle is challenging for the modern ocean and even more so for the poorly constrained LGM paleo-environment. As an alternative approach, we query the carbon isotopic record to identify what mechanisms controlled the glacial-interglacial $CO_2$ swings.

On the timescales at which Quaternary deglaciations occur (~5000 years), two main carbon reservoirs have a significant exchange with the atmosphere: organic carbon stored on land, and dissolved inorganic carbon (DIC) stored in the ocean. Proxies and vegetation models both suggest that carbon in terrestrial biomass increased during the deglaciation[15–17], making the land a sink that would have resulted in a ~20 ppmv atmospheric $CO_2$ decrease[18]. The available evidence thus points toward the ocean as the source of the deglacial increase in atmospheric $CO_2$.

[1]Department of Earth, Environmental, and Planetary Sciences, Case Western Reserve University, Cleveland, OH, USA. [2]School of Environmental Sciences, University of Liverpool, Liverpool, UK. [3]Department of Earth, Atmospheric, and Planetary Sciences, Massachusetts Institute of Technology, Cambridge, MA, USA. ✉e-mail: anne.omta@case.edu

Carbon storage in the ocean is primarily set by its solubility in water as determined by temperature, salinity and total alkalinity (a measure of acid buffering capacity). The solubility of $CO_2$ in water decreases as the temperature and salinity rise. On one hand, the ~3 °C increase in average ocean temperature from the LGM to the Holocene would have resulted a ~30 ppmv atmospheric $CO_2$ increase[19]. On the other hand, the decrease in ocean salinity associated with the melting of most Northern Hemisphere ice sheets would have decreased atmospheric $CO_2$ by ~6 ppmv[20]. Thus, accounting for the relatively well constrained changes in terrestrial biomass, ocean temperature, and salinity leaves essentially the entire ~90 ppmv of the atmospheric $CO_2$ rise unexplained. Here, we investigate the hypothesis that marine productivity played a key role in the LGM-to-Holocene $CO_2$ change.

Marine photosynthesizers grow by taking up DIC and nutrients in the euphotic layer of the ocean. When they die, part of their biomass sinks into deeper waters where it is consumed by organisms such as bacteria. Eventually, the organic carbon is remineralised into DIC through respiration at different trophic levels. As a result of this biological carbon pump, the DIC concentration is higher in the deep ocean than at the surface[21]. In upwelling regions such as the Southern Ocean, carbon-rich deep waters come in contact with the atmosphere. This leads to outgassing from the ocean to the atmosphere, since these waters are oversaturated in carbon. In the present climate the oversaturation is not completely erased, because the equilibration process takes ~1 year, which is of the same order as the surface residence time of the water in the Southern Ocean. As a result, a significant fraction of upwelled carbon of biological origin is resubducted into the ocean interior, before it has fully equilibrated with the atmosphere[22]. This disequilibrium DIC increases the amount of carbon stored in the ocean, in addition to carbon storage due to sinking of organic matter[11,13,23]. Disequilibrium DIC can also have a physical origin as is the case in the North Atlantic. Here the cooling of Northward-flowing surface water results in a negative disequilibrium, i.e., the DIC of the surface waters is lower than the saturated value. This has been termed the physical disequilibrium DIC as opposed to the biological disequilibrium DIC[12] that we will be focusing on.

Using proxy reconstructions of carbonate ion concentrations at the LGM, Goodwin & Lauderdale[24] estimated a $55 \pm 15$ ppmv storage of carbon in the ocean at the LGM, but they could not distinguish between the relative contributions of soluble, respired, or disequilibrium DIC. Recently, using deep-ocean oxygen concentration reconstructions at the LGM[25,26], Vollmer et al.[27] inferred that a more efficient LGM biological pump accounted for a $64 \pm 28$ ppmv atmospheric $CO_2$ drawdown before carbonate compensation, i.e., the stabilization of $CO_2$ levels through the dissolution of calcium carbonate in the ocean. However, any estimate of carbon storage based on oxygen is complicated by $CO_2$'s much longer air-sea equilibration time (~1 year) compared to oxygen (~1 month). To assess the potential bias in this estimate of biologically sequestered carbon in the ocean, we perform a similar calculation using stable carbon isotopic ratios ($\delta^{13}C$) instead of oxygen proxies. We focus on the vertical $\delta^{13}C$ gradient in the ocean, which provides another measure of sequestered carbon[28–31]. Since $\delta^{13}C$ has a much longer equilibration timescale (~10 years) than $CO_2$[32,33], we expect the bias in this estimate to be in the opposite direction compared to the oxygen-based estimate in[27]. Thus, our analysis provides an independent constraint on how much of the atmospheric $CO_2$ change can be attributed to a change in biologically sequestered carbon in the ocean.

Many physical, chemical, and biological processes involve isotopic fractionation, which means that they select for one isotope of an element over a different isotope of that same element. Therefore, isotopic ratios can serve as proxies for the underlying processes. Here we focus on carbon, which exists in two stable isotopic forms: $^{12}C$ (light carbon) and $^{13}C$ (heavy carbon). The isotopic composition of carbon in the atmosphere and ocean is typically quantified in terms of the ratio of the heavy to light stable isotopes, $R = \frac{^{13}C}{^{12}C}$. Since $R$ is very small ($\simeq 0.011$) and we care about deviations from its standard value ($R_s$), it is common practice to use the quantity $\delta^{13}C = \frac{R - R_s}{R_s} \times 1000‰$. The value of $\delta^{13}C$ in past atmospheres is measured from gas bubbles trapped in ice cores. The ocean values are inferred from the isotopic composition of calcium carbonate shells of benthic foraminifera buried in sediments.

On average, $\delta^{13}C$ is nearly 7‰ higher in the ocean than in the atmosphere. This overall fractionation is the result of two processes. First, isotopic fractionation associated with air-sea gas exchange leads to the $\delta^{13}C$ of surface ocean DIC being ~8–10‰ higher than the atmosphere[34]. Second, the photosynthetic process has a preference for $^{12}C$ over $^{13}C$. As a result, the $\delta^{13}C$ of organic carbon is on average ~25‰ lower than surface ocean DIC[30]. Sinking and respiration of organic matter then adds some of this isotopically light carbon to the deep ocean, making the average $\delta^{13}C$ of ocean DIC slightly lower than its surface value. Thus, biological activity decreases the difference in $\delta^{13}C$ between the atmosphere and the ocean average.

In Fig. 1, we show the evolution of: (a) atmospheric $pCO_2$ based on Antarctic ice cores[35]; and (b) $\delta^{13}C$ over the last 20,000 years for the atmosphere, based on Antarctic ice cores[36], as well as $\delta^{13}C$ for the global ocean average, based on a compilation of 127 sediment cores from the different oceans (data details and core locations can be found in[31]). Atmospheric $\delta^{13}C$ is a proxy for the distribution of carbon across different reservoirs, rather than the amount of carbon in the atmosphere[37]. As such, it behaved rather differently from atmospheric $pCO_2$ across the deglaciation from the LGM into the Holocene. While atmospheric $pCO_2$ increased almost monotonically from 18 to 11 kyr BP, atmospheric $\delta^{13}C$ first decreased and then increased. The sharp drop between 17 and 16 kyr BP has been attributed to the outgassing of isotopically light carbon from the ocean[36] and the release of terrestrial carbon[38]. From 12 kyr BP until 6 kyr BP, atmospheric $\delta^{13}C$ increased at approximately the same rate as the average ocean $\delta^{13}C$. This has been interpreted as a result of the regrowth of boreal forests[36]. However, caution should be exercised when interpreting such transient deglacial $\delta^{13}C$ changes in terms of mass fluxes. Deglaciations are, by definition, times of rapid climatic change and an unstable global carbon cycle. For example, the thawing permafrost and retreating ice sheets may have led to the oxidation of organic matter[39] or even of exhumed petrogenic carbon[40]. Thus, isotopically light carbon from reservoirs other than the ocean would have been released to the atmosphere, temporarily increasing the ocean-atmosphere $\delta^{13}C$ difference. Eventually, this temporary increase would have been erased by re-equilibration between the ocean and the atmosphere. In other words, the ocean-atmosphere $\delta^{13}C$ difference provides a more unambiguous measure of biologically sequestered carbon in the ocean for times with a relatively stable climate, such as the LGM and Holocene, compared to periods of rapid change, such as the deglaciation. As the absolute changes in the size of other mobile carbon stores (such as terrestrial biomass) are small relative to the entire ocean-atmosphere carbon inventory, the effect of these reservoirs on the total $CO_2$ change after equilibration must also be small. By focusing on changes in the overall difference between atmospheric and deep ocean $\delta^{13}C$ during periods of slow change (the LGM and the Holocene), we link the vertical gradients in isotopic composition directly to changes in quasi-steady state carbon storage in the sea.

Based on the impact of temperature on physical fractionation alone, one would expect the difference between the $\delta^{13}C$ of the ocean and the atmosphere to have decreased from the LGM to the Holocene. The opposite is observed: the difference in $\delta^{13}C$ between the ocean average and the atmosphere increased from $6.4 \pm 0.1‰$ at the LGM to $6.7 \pm 0.1‰$ during the Holocene. This is consistent with a larger inventory of biologically sequestered carbon in the ocean at the LGM than during the Holocene, which we quantify in the next Section.

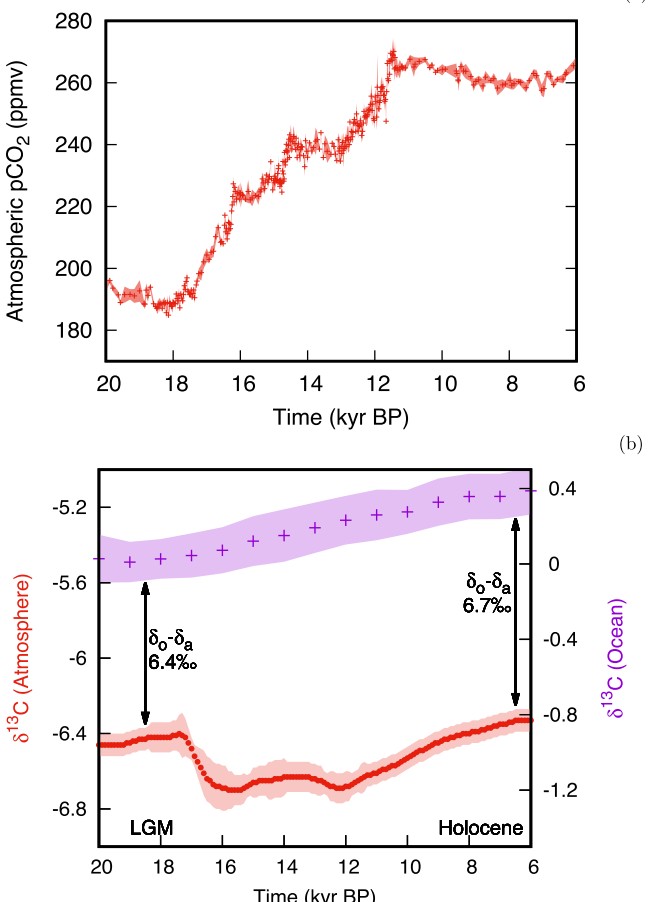

**Fig. 1 | Atmospheric CO₂ and atmospheric and oceanic $\delta^{13}$C exhibit distinct patterns across the last deglaciation. a** Measured atmospheric pCO₂ (in red, reproduced from ref. 35) from 20 to 6 kyr before present. **b** Measured atmospheric (red, left vertical axis, reproduced from ref. 36) and ocean-average (purple, right vertical axis, reproduced from ref. 31) $\delta^{13}$C from 20 to 6 kyr before present. The vertical arrows indicate the larger $\delta^{13}$C difference between the ocean ($\delta_o$) and atmosphere ($\delta_a$) during the Holocene than at the Last Glacial Maximum. A map with the locations of the 127 core sites used for the ocean-average $\delta^{13}$C was provided in ref. 31. In both panels, shaded areas indicate 2-$\sigma$ uncertainty intervals of the measurements. Note that the vertical axes have different scales and that time runs forward to the right.

## Results

This section sketches our budget calculations, with the details provided in the "Methods" section. We formulate a budget for $\delta^{13}$C that we convert into a carbon budget using the connections between isotopic fractionation and carbon cycle processes. More specifically, we use the known fractionation associated with air-sea equilibration to estimate its contribution to the overall observed LGM-Holocene $\delta^{13}$C changes. Then, by placing reasonable boundaries on the disequilibrium component of the fractionation, we calculate a residual change $\delta^{13}$C that we attribute to a deglacial decrease in the biologically sequestered carbon inventory.

### Ocean carbon reservoirs

To understand the impacts of different processes, it is useful to divide the ocean carbon into saturated, disequilibrium, and regenerated components[41,42]. The saturated carbon is defined as the DIC concentration that a water mass would have under perfect air-sea equilibration just before it leaves the surface to sink into the ocean interior. After subduction, a water mass collects sinking organic material that is oxidized by deep-sea organisms such as bacteria. The regenerated

carbon is defined as the amount of DIC added to a water mass through this process. The disequilibrium carbon is then the difference between the actual DIC concentration and the sum of the saturated and regenerated carbon. At the ocean surface, the disequilibrium carbon concentration is equal to the actual air-sea carbon disequilibrium (regenerated carbon is equal to zero at the surface by definition).

Figure 2 illustrates the role of the global ocean overturning circulation in the ocean-atmosphere partitioning of carbon, $\delta^{13}$C, and oxygen throughout the deep ocean. The overturning circulation consists of two main components: the surface circulation (upper ~1 km, not shown in the figure) and the deep circulation (below ~1 km depth). The deep circulation sketched in Fig. 2 consists of sinking water into the deep ocean through convection near Greenland and Antarctica, as well as through upwelling of deep water across various regions of the World Ocean. Deep water formed near Greenland flows southward and fills most of the deep Atlantic Ocean, as represented by the blue line in Fig. 2A. Deep water formed near Antarctica fills the deep Indian and Pacific Oceans, as represented by the black line in Fig. 2A. The two branches come to the surface in the Southern Ocean where they connect resulting in a figure-eight global overturning loop[43]. Due to the relatively long residence of surface water in the Atlantic, deep water forming near Greenland (vertical blue line) is approximately equilibrated with the atmosphere in terms of oxygen and carbon[44]. As this deep water flows southward in the ocean interior (lower horizontal blue line), it collects regenerated carbon when sinking organic particles are respired back to their inorganic constituents. In this process, oxygen is progressively depleted and the $\delta^{13}$C of the water mass decreases because of the particles' low $\delta^{13}$C value (due to isotopic fractionation during photosynthesis). Water enriched in regenerated carbon continues South until it upwells in the Southern Ocean (slanted blue line), which gives rise to a large air-sea DIC disequilibrium equivalent to the actual air-sea pCO₂ difference. As regenerated carbon is equal to zero at the surface by definition, regenerated carbon is converted into (relabeled as) disequilibrium carbon once the deep water reaches the surface. This disequilibrium is partly, but not entirely, eroded before the water subducts near Antarctica due to the relatively short 1-year residence time of surface water in the Southern Ocean[22,44].

We will be using $m_{sat}$, $m_{reg}$, and $m_{dis}$ for the total oceanic inventories of saturated, regenerated, and disequilibrium carbon, respectively. The total carbon inventory, $m$, is thus the sum of these three components:

$$m = m_{sat} + m_{reg} + m_{dis} \qquad (1)$$

We define the sum of the regenerated and disequilibrium carbon as the (biologically) sequestered carbon $m_{seq}$, similar to[45]:

$$m_{seq} = m_{reg} + m_{dis} \qquad (2)$$

We are using the term biologically sequestered carbon, because the disequilibrium carbon has a primarily biological origin. This does not imply that the sequestration is driven solely by biology, as sea ice and the ocean circulation play key roles in maintaining the carbon disequilibrium.

### Bulk biologically sequestered carbon budget

Now we divide the ocean-average $\delta^{13}$C into contributions from saturated and sequestered carbon ($\bar{\delta}_{o,sat}$ and $\bar{\delta}_{o,seq}$). These are different from $m_{sat}$ and $m_{seq}$, which correspond to carbon inventories. Furthermore, we wish to emphasize that the behavior of $m_{seq}$ and $\bar{\delta}_{seq}$ is somewhat different due to the different equilibration times of DIC and $\delta^{13}$C[33]. We indicate the atmospheric $\delta^{13}$C as $\bar{\delta}_a$.

Isotopic fractionation is associated with air-sea gas exchange and with photosynthesis. The $\delta^{13}$C difference between the saturated carbon

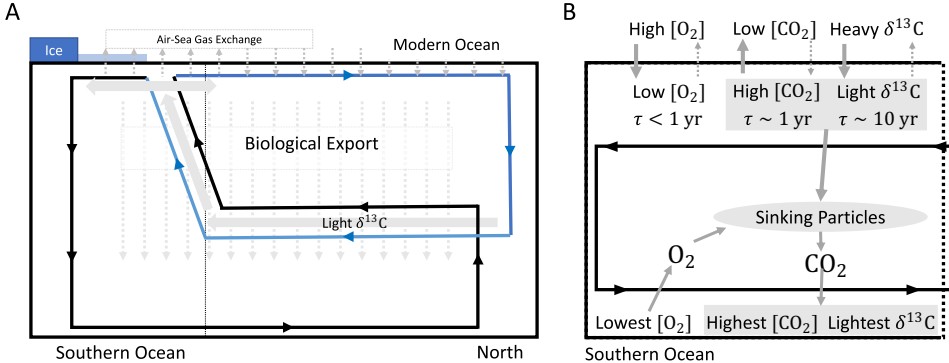

**Fig. 2 | A schematic depiction of the role of the oceanic overturning circulation in the fluxes of carbon and oxygen between the atmosphere and ocean. A** The paths of the Northern (blue) and Southern (black) overturning cells for the modern ocean with the processes that move carbon and isotopes in the sea (air-sea gas exchange, biological export, and flow). **B** A zoom-in on the Southern portion of the Southern overturning cell shows how oxygen, carbon, and carbon isotopes interact with the atmosphere and sinking particles to generate disequilibrium.

and the atmosphere, $\bar{\epsilon}_d = \bar{\delta}_{o,sat} - \bar{\delta}_a$, is given by the equilibrated air-sea fractionation ($\simeq 10‰$)[34]. The $\delta^{13}C$ difference between the saturated and the sequestered carbon, $\bar{\epsilon}_{seq} = \bar{\delta}_{o,sat} - \bar{\delta}_{o,seq}$, is determined by photosynthetic fractionation and by air-sea equilibration through the following mechanism. Water enriched in isotopically light regenerated carbon upwells in the Southern Ocean, giving rise to a disequilibrium between the atmosphere and the ocean surface both in terms of DIC and $\delta^{13}C$. DIC has a ~1 year air-sea equilibration time, which is similar to the residence time of water at the surface of the Southern Ocean. Therefore, the DIC disequilibrium is partly erased by air-sea gas exchange before the water is subducted. In contrast, $\delta^{13}C$ has a significantly longer air-sea equilibration time (~10 years) than the surface residence time of water in the Southern Ocean. As a result, the $\delta^{13}C$ disequilibrium is largely retained at subduction. This has a somewhat counterintuitive implication for the isotopic bookkeeping. Consider a water mass upwelling and subducting in the Southern Ocean. For this water mass, we can write a carbon isotopic budget: $\delta_w \simeq \delta_{w,sat} - \frac{C_{w,seq}}{C_{w,sat}}\epsilon_{w,seq}$. Here, $\delta_w$ is the $\delta^{13}C$ value of the water mass, $\delta_{w,sat}$ is the saturated $\delta^{13}C$ at the surface of the Southern Ocean, $C_{w,seq}$ is the biologically sequestered carbon in the water mass, $C_{w,sat}$ is the saturated DIC concentration at the surface of the Southern Ocean, and $\epsilon_{w,seq}$ is the $\delta^{13}C$ difference between the saturated and sequestered DIC in the water mass. As the water mass travels along the surface of the Southern Ocean, $\delta_w$ remains approximately constant because of the long timescale for isotopic equilibration (~10 years). Furthermore, $\delta_{w,sat}$ (the saturated $\delta^{13}C$) is unaffected by air-sea gas exchange. However, air-sea gas exchange does lead to a decrease in the ratio of sequestered carbon to saturated DIC in the surface ocean ($\frac{C_{w,seq}}{C_{w,sat}}$). To maintain isotopic mass balance, $\epsilon_{w,seq}$ then has to increase along the path of the water parcel at the surface of the Southern Ocean. This isotopic offset enhanced by air-sea gas exchange is retained after the water mass is subducted and traveling through the deep ocean. As such, $\bar{\epsilon}_{seq}$ is not a true fractionation, which is why we refer to it as the sequestered offset factor. In the "Methods" section, we use chemical and isotopic data to estimate that $\bar{\epsilon}_{seq} \simeq 40‰$ during the Holocene.

The LGM-to-Holocene change in the ocean-atmosphere $\delta^{13}C$ difference can now be divided into contributions from physical air-sea fractionation and from the sequestered carbon pool:

$$\Delta\bar{\delta}_o - \Delta\bar{\delta}_a \simeq \Delta\bar{\epsilon}_d - \Delta\left(\frac{m_{seq}}{m}\bar{\epsilon}_{seq}\right) \qquad (3)$$

where $\bar{\delta}_o$ and $\bar{\delta}_a$ represent the average $\delta^{13}C$ in the ocean and atmosphere respectively and the $\Delta$ operator represents the difference between the Holocene and LGM values. We take $\Delta\bar{\delta}_o = 0.32 \pm 0.10‰$

(1-$\sigma$ uncertainty)[46] and $\Delta\bar{\delta}_a = 0.10 \pm 0.10‰$ [47]. Taking the LGM-to-Holocene change in average ocean temperature equal to $2.57 \pm 0.24\,°C$[48] and the pCO$_2$ change equal to 90 ppmv[35], we estimate $\Delta\bar{\epsilon}_d = -0.18 \pm 0.03‰$ (see "Methods" section). Substituting these values in eq. (3), we then estimate:

$$\Delta\left(\frac{m_{seq}}{m}\bar{\epsilon}_{seq}\right) \simeq \Delta\bar{\epsilon}_d - \Delta\bar{\delta}_o + \Delta\bar{\delta}_a \simeq -0.4 \pm 0.2‰ \qquad (4)$$

To gain insight into what specific process drove this change, we make a first-order Taylor expansion of the terms in the parenthesis (see Methods for more details):

$$\underbrace{\left(\bar{\epsilon}_{seq}\frac{m_{seq}}{m}\right)}_{\text{Prefactor}}\left(\underbrace{\frac{\Delta m_{seq}}{m_{seq}}}_{\text{Seq.C}} - \underbrace{\frac{\Delta m}{m}}_{\text{TotalC}} + \underbrace{\frac{\Delta\bar{\epsilon}_{seq}}{\bar{\epsilon}_{seq}}}_{\text{Fractionation}}\right) \simeq -0.4 \pm 0.2‰. \qquad (5)$$

The prefactor of eq. (5) is undoubtedly positive, which implies that at least one of the three terms inside the second set of parentheses ($\frac{\Delta m_{seq}}{m_{seq}}$, $-\frac{\Delta m}{m}$, $\frac{\Delta\bar{\epsilon}_{seq}}{\bar{\epsilon}_{seq}}$) be negative. These terms represent relative changes: in the ocean's sequestered component; in the ocean's total carbon; and in the fractionation of sequestered carbon in the ocean. The detailed quantification of each term in eq. (5) is presented in the "Methods" section. Here we summarize the key results.

During deglaciation, carbon is transferred from the ocean to the atmosphere[13,20,49]. This decreases the total ocean carbon inventory, which means that the second term is positive. The Holocene-LGM difference in the sequestered isotopic offset ($\Delta\bar{\epsilon}_{seq}$) can be estimated by realizing that $\bar{\epsilon}_{seq}$ is always larger than ~25‰ (the photosynthetic fractionation) and that the LGM value of $\bar{\epsilon}_{seq}$ must have been lower than the Holocene value of ~40‰ for two main reasons. Firstly, the expanded sea-ice cover at the LGM would have led to an increase in the disequilibrium carbon. Secondly, $\bar{\epsilon}_{seq}$ increases with the ratio of the equilibration times of $\delta^{13}C$ and DIC. This ratio was likely smaller at the LGM than during the Holocene, as explained in the "Methods" section. Thus, we have lower and upper bounds of 0 and 15‰ for $\Delta\bar{\epsilon}_{seq}$, which represent unlikely extremes. $\Delta\bar{\epsilon}_{seq} = 0‰$ requires air-sea gas exchange to be larger at the LGM, when taking into account that pCO$_2$ was lower at LGM than during the Holocene. $\Delta\bar{\epsilon}_{seq} = 15$ requires the absence of any gas exchange in the Southern Ocean at the LGM. As we don't have any further information about the probability distribution of $\Delta\bar{\epsilon}_{seq}$, we estimate $\Delta\bar{\epsilon}_{seq} = 7.5 \pm 7.5‰$ (with the central values more likely than the extreme ones).

Using the Holocene values for $\bar{e}_{seq} \simeq 40‰$, $m_{seq} \simeq 0.08 \times 10^{18}$ mol, and $m \simeq 3 \times 10^{18}$ mol[50], we then find:

$$\Delta m_{seq} \simeq (-0.08 \pm 0.05) \times 10^{18} \text{mol} \qquad (6)$$

Converting moles to petagrams, this amounts to a decrease of $1000 \pm 600$ Pg in biologically sequestered carbon for the Holocene compared to the LGM. Before equilibration of the ocean alkalinity budget through the process of carbonate compensation[32], this implies a Holocene-LGM increase in $pCO_2 \simeq 65 \pm 35$ ppmv. After accounting for a change in ocean alkalinity associated with carbonate compensation, the change increases to $pCO_2 \simeq 75 \pm 40$ ppmv. Hence, the inferred $pCO_2$ increase could be sufficient to explain the entire ~90 ppm glacial-interglacial difference (within uncertainty).

## Mechanisms of biological carbon sequestration

What are the relative contributions of the regenerated and disequilibrium carbon to the LGM to Holocene difference in biologically sequestered carbon? We provide a detailed calculation in the "Methods" section and here summarize the key results. Consider $\delta^{13}C$ near deep-water formation regions in the North Atlantic and Southern Ocean[51]. These waters start their journey at the surface with no regenerated carbon, accumulating regenerated carbon as they flow through the abyss. As a result, $\delta^{13}C$ data near deep-water formation sites provide an estimate of the preformed $\delta^{13}C$ of a water mass before it is altered by organic matter respiration. In contrast, the ocean-averaged $\delta^{13}C$ includes the globally averaged impact of organic matter respiration. By subtracting the ocean-averaged preformed $\delta^{13}C$ from the ocean-averaged $\delta^{13}C$, we infer the regenerated component.

We estimate the ocean-averaged preformed $\delta^{13}C$ by summing the preformed $\delta^{13}C$ of Northern- and Southern-sourced waters, weighted by their respective fractions of the global ocean volume. There has been debate about these water-mass fractions at the LGM[52]. Sharp vertical gradients in $\delta^{13}C$, Cd/Ca, and $\delta^{18}O$ in the Atlantic at the LGM have been interpreted as evidence of a boundary between Northern- and Southern-sourced water at ~2000 m[53–55]. This would imply that Northern-sourced waters filled ~10% of the deep ocean at the LGM, compared to ~40% during the Holocene[56]. Paleo-reconstructions based on Neodymium isotopes instead suggest little change in the water-mass configuration between the LGM and the Holocene[57,58]. To account for this range of possible LGM water-mass configurations, we estimated the change in regenerated carbon for a range of fractional changes in Northern-sourced waters from the LGM to Holocene $\Delta\alpha$ between 0-30% shown in Fig. 3. We found that the estimated change in the regenerated carbon ($\Delta m_{reg}$) becomes a progressively smaller fraction of the sequestered carbon ($\Delta m_{seq}$) for increasing $\Delta\alpha$.

$\Delta\alpha = 0$ corresponds to the limit of no LGM to Holocene change in the water-mass configuration. In this limit, the regenerated component accounts for between −20% and +110% (2-$\sigma$ uncertainty) of the sequestered carbon change, with a most likely value of +50%. In other words, our analysis is inconclusive about the contribution of regenerated carbon to the change in sequestered carbon in this regime. $\Delta\alpha = 0.3$ instead corresponds to the limit of Northern-sourced water being confined to the upper 2000 m of the Atlantic at the LGM. In this limit, the regenerated component accounts for between −40% and +50% (2-$\sigma$ uncertainty) of the sequestered carbon change, with a most likely value of +7%. In other words, regenerated carbon accounts for no more than half of the change in sequestered carbon in this regime.

While both $\Delta\alpha$ limits have some support in paleo-proxy data, they imply very different ocean circulations and biological disequilibrium at the LGM. Ocean dynamics suggest that if the volume occupied by Northern-sourced waters did not change ($\Delta\alpha = 0$), then the overturning strength would likely not have changed appreciably either[59,60]. Furthermore, simulations have suggested that with the current overturning circulation, increased sea-ice coverage increases

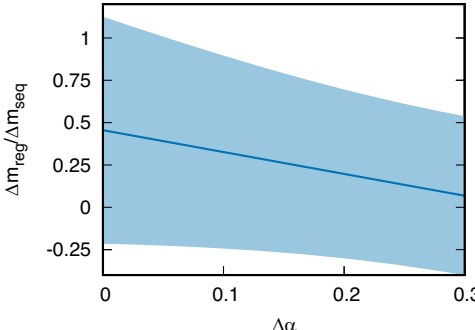

**Fig. 3 | Estimated change in the regenerated carbon as a fraction of the change in the sequestered carbon ($\frac{\Delta m_{reg}}{\Delta m_{seq}}$) as a function of the assumed change in the fraction of Northern-sourced water ($\Delta\alpha$).** $\Delta\alpha = 0$ corresponds to no LGM to Holocene change in the water-mass configuration, whereas $\Delta\alpha = 0.3$ corresponds to Northern-sourced water being confined to the upper 2000 m of the Atlantic at the Last Glacial Maximum. The shaded area indicates the 2-$\sigma$ uncertainty interval of the estimate. Source data and the Matlab code to generate this figure are provided as Source Data files in the online Supplementary Material.

disequilibrium DIC by only a modest amount[12]. Therefore, changes in ocean carbon storage would indeed have to be associated with $\Delta m_{reg}$ rather than $\Delta m_{seq}$, consistent with our analysis. The $\Delta\alpha = 0.3$ limit implies a contraction of the volume of Northern-sourced waters at the LGM, which could have been forced by the expanded sea ice[43]. The expanded sea ice over the expanded Southern overturning cell could have provided an effective blocking of air-sea gas exchange. This would in turn be consistent with our inference that the change in sequestered carbon was mostly due to the disequilibrium component in this limit. In the next section we argue that independent $\Delta^{14}C$ and oxygen proxy data are consistent with the second limit, but not the first.

## Discussion

Our carbon isotopic mass budget suggests that the increase in atmospheric $CO_2$ from the LGM to the Holocene was driven by a decrease in biologically sequestered carbon in the ocean. This calculation used the assumption that the biologically mediated difference in $\delta^{13}C$ between the atmosphere and the surface of the Southern Ocean was either comparable or smaller at the LGM than during the Holocene. Thus, the Southern Ocean carbon must have been similarly or more poorly equilibrated with the atmosphere at the LGM compared to the Holocene. This seems plausible in view of the documented expansion of Antarctic sea ice[61] and the expected strengthening of the overturning circulation in response to an increased surface buoyancy loss[52] at the LGM. Should new evidence point to a weakened LGM overturning circulation, then our $\delta^{13}C$ budget should also be revised. This said, the impact of the overturning strength on the $\delta^{13}C$ and carbon budgets is likely relatively minor due to counteracting effects. For example, a stronger LGM circulation may not only decrease air-sea equilibration but also nutrient utilization, which would increase the disequilibrium.

Our $\delta^{13}C$-based estimate of the LGM to Holocene change in $pCO_2$ before carbonate compensation ($65 \pm 35$ ppmv) is in close agreement with estimates based on $\Delta^{14}C$ ($50 \pm 27$ ppmv[62]) and oxygen ($64 \pm 28$ ppmv[27]). A key assumption underlying the use of $\Delta^{14}C$ to infer changes in sequestered carbon is that export production rates have been approximately constant. The similarity between the estimate based on $\Delta^{14}C$ and those based on $\delta^{13}C$ suggests that this assumption is justified. Constant export production is in turn consistent with most of the change in biologically sequestered carbon being due to DIC disequilibrium. Thus, the $\Delta\alpha = 0.3$ regime (Northern-sourced water confined to upper 2000 m of Atlantic at LGM) provides the most internally consistent picture. The consistency between the

oxygen- and $\delta^{13}C$-based estimate is significant as well because the two proxies suffer from opposite biases. Consider the schematics in Fig. 2, which depict the role of the modern oceanic overturning circulation in regulating the air-sea fluxes of $CO_2$, oxygen, and $\delta^{13}C$. Water upwelling in the Southern Ocean is not in equilibrium with the atmosphere. Due to organic matter respiration, it is undersaturated in oxygen, oversaturated in DIC, and its $\delta^{13}C$ is below its saturated value. Once at the surface, the water spends about 1 year in contact with the atmosphere before being resubducted. This time is comparable to the air-sea equilibration timescale for $CO_2$[32], but much longer than that of oxygen (~1 month[32]) and much shorter than that of $\delta^{13}C$ (~10 years). Thus, the deviation from equilibrium in newly formed deep water is smaller for oxygen than for carbon, while the opposite it true for $\delta^{13}C$. As a result, the total biologically sequestered carbon may be underestimated by oxygen utilization and overestimated by $\delta^{13}C$. The agreement between the oxygen- and $\delta^{13}C$-based estimates can be explained with a reduction in Southern Ocean air-sea gas exchange at the LGM compared to the Holocene. If air-sea fluxes were indeed inhibited at the LGM, not only the disequilibrium in $\delta^{13}C$ and carbon but also the disequilibrium in oxygen would have been largely retained at resubduction. Under this scenario, oxygen utilization more closely represents the total biologically sequestered carbon reservoir, as estimated from $\delta^{13}C$, rather than the regenerated carbon alone. We estimate (see "Methods" section) that this would imply a Holocene-LGM difference in global ocean average disequilibrium oxygen $\Delta O_{2,dis} \simeq -80 \pm 50\,\mu M$. This is broadly consistent with the results from a recent modeling experiment (see Fig. 8B, C in ref. 42). Indeed, proxy data suggest decreased Southern Ocean air-sea gas exchange at the LGM compared to the Holocene[63,64].

Our $\delta^{13}C$-based estimate of the LGM-to-Holocene change in biologically sequestered carbon in the ocean translates into a $pCO_2$ change of $75 \pm 40$ ppmv after carbonate compensation. This could explain the entire Holocene-LGM difference, since the observed 90 ppmv lies within the estimated uncertainty interval. That said, it is also possible that biologically sequestered carbon in the ocean does not account for the full 90 ppmv of $CO_2$ change. Given the good correspondence between the estimates based on $O_2$ and $\delta^{13}C$, any remaining unexplained $CO_2$ change would likely be due to a factor that has minimal impacts on either quantity. One such factor is whole-ocean alkalinity, which was likely higher at the LGM[65,66] and would thus have generated a further decrease of atmospheric $CO_2$[67,68]. However, the magnitude of this effect is difficult to quantify as we lack direct paleo-proxies for alkalinity.

The main contribution of our work is to provide data-based constraints on the LGM carbon and $\delta^{13}C$ budgets that have primarily been studied through the lens of numerical simulations[5,11,12,69–73]. Simulations rely on assumptions in the model formulations, for example with respect to changes in air-sea gas exchange and carbon export. Our data-based calculations suggest that differences in Southern Ocean carbon disequilibrium played a major role in glacial-interglacial $CO_2$ changes, which narrows down the range of potential processes. We hope that the constraints emerging from our analysis provide a useful test for coupled models of global climate and the carbon cycle. Ultimately, this should improve model predictions of changes in the carbon cycle in both paleo and modern contexts.

## Methods

In this Section, we derive the carbon budget that we used in the main text to infer the changes in atmospheric carbon between the LGM and Holocene. More specifically, we first estimate the Holocene-LGM difference in $m_{seq}$ and its impact on atmospheric $pCO_2$ (Section "Sequestered carbon mass budget"). To estimate the individual contributions from $m_{dis}$ and $m_{reg}$ toward $m_{seq}$, we calculate the Holocene-LGM $m_{reg}$ difference in Section "Regenerated carbon budget". In Section "Dependence of equilibrated air-sea isotopic

**Table 1 | Definition of key quantities with associated units**

| Symbol | Description | Units |
|---|---|---|
| $R$ | Measured $^{13}C/^{12}C$ ratio | – |
| $R_s$ | Standard $^{13}C/^{12}C$ ratio | – |
| $\delta^{13}C$ | Deviation of measured $^{13}C$ from standard | ‰ |
| $m$ | Total amount of carbon in the ocean | mol |
| $m_{sat}$ | Total saturated carbon | mol |
| $m_{seq}$ | Total sequestered carbon | mol |
| $C_{seq}$ | Ocean average sequestered carbon concentration | mol/m³ |
| $\bar{\delta}_o$ | Mass-averaged $\delta^{13}C$ in the ocean | ‰ |
| $\bar{\delta}_a$ | Atmospheric $\delta^{13}C$ | ‰ |
| $\bar{\delta}_{o,sat}$ | Saturated $\delta^{13}C$ | ‰ |
| $\bar{\delta}_{o,seq}$ | Sequestered $\delta^{13}C$ | ‰ |
| $\bar{\epsilon}_d$ | Air-sea carbon isotopes fractionation factor | ‰ |
| $\bar{\epsilon}_p$ | Photosynthetic carbon isotopes fractionation factor | ‰ |
| $\bar{\epsilon}_{seq}$ | Sequestered isotopic offset factor | ‰ |

fractionation on atmospheric $CO_2$", we focus on the impact of atmospheric $CO_2$ on the air-sea isotopic fractionation. Table 1 provides a list of key quantities used in these calculations.

### Sequestered carbon mass budget

To formulate our carbon isotope mass budgets, we introduce the mass-averaged ocean $\delta^{13}C$:

$$\bar{\delta}_o = \frac{1}{m} \int \delta_o \, dm \qquad (7)$$

For our sequestered carbon mass budget, we divide the integral in eq. (7) into saturated and sequestered carbon components:

$$\bar{\delta}_o \simeq \frac{m_{sat}}{m} \bar{\delta}_{o,sat} + \frac{m_{seq}}{m} \bar{\delta}_{o,seq} \simeq \bar{\delta}_{o,sat} - \frac{m_{seq}}{m} \left( \bar{\delta}_{o,sat} - \bar{\delta}_{o,seq} \right) \qquad (8)$$

where $\bar{\delta}_{o,seq}$ and $\bar{\delta}_{o,sat}$ are the respective mass-averaged $\delta^{13}C$ values. Since our goal is to compare the changes in ocean $\bar{\delta}_o$ to those in the atmosphere, $\bar{\delta}_a$, we focus on the difference

$$\bar{\delta}_o - \bar{\delta}_a \simeq \left( \bar{\delta}_{o,sat} - \bar{\delta}_a \right) - \frac{m_{seq}}{m} \left( \bar{\delta}_{o,sat} - \bar{\delta}_{o,seq} \right) \qquad (9)$$

The first term in brackets on the right-hand side of eq. (9) equals the physical fractionation between the atmosphere and the saturated carbon $\bar{\epsilon}_d$ ($\simeq 10$‰):

$$\bar{\epsilon}_d = \bar{\delta}_{o,sat} - \bar{\delta}_a \qquad (10)$$

We define the second bracketed term in eq. (9) as the sequestered isotopic offset factor $\bar{\epsilon}_{seq}$:

$$\bar{\epsilon}_{seq} = \bar{\delta}_{o,sat} - \bar{\delta}_{o,seq} \qquad (11)$$

Combining eqs. ((9)–(11)) yields:

$$\bar{\delta}_o - \bar{\delta}_a \simeq \bar{\epsilon}_d - \frac{m_{seq}}{m} \bar{\epsilon}_{seq} \qquad (12)$$

For the Holocene, we have sufficient data for a rough estimate of $\bar{\epsilon}_{seq}$: First, consider that $\bar{\delta}_o \simeq 0.5$‰[74], $\bar{\delta}_a \simeq -6.3$‰[36], $\bar{\epsilon}_d \simeq 10$‰[34] (assuming a mean ocean temperature of 3 °C). Furthermore, the mean ocean DIC concentration $\bar{C} = 2.30$ mol m⁻³[21] and the mean ocean DIC

concentration $\bar{C}_{sat} = 2.12$ mol m$^{-3}$ (at 3 °C)[75], which gives $\bar{C}_{seq} = \bar{C} - \bar{C}_{sat} = 0.18$ mol m$^{-3}$ and thus $\frac{m_{seq}}{m} = \frac{C_{seq}}{C} = \frac{0.18}{2.30}$. Substituting these values into eq. (12) and rearranging, we obtain $\bar{\epsilon}_{seq} \simeq 40‰$.

To indicate differences between Holocene minus LGM values we introduce the symbol $\Delta$, e.g., $\Delta\bar{\delta}_o = \bar{\delta}_o^{Holo} - \bar{\delta}_o^{LGM}$. The budget in (12) then gives:

$$\Delta\bar{\delta}_o - \Delta\bar{\delta}_a \simeq \Delta\bar{\epsilon}_d - \Delta\left(\frac{m_{seq}}{m}\bar{\epsilon}_{seq}\right) \quad (13)$$

with $\Delta\bar{\delta}_o \simeq 0.32 \pm 0.10‰$ (1-$\sigma$ uncertainty)[46] and $\Delta\bar{\delta}_a \simeq 0.10 \pm 0.10‰$[47]. Eq. (13) thus implies that $\Delta\left(\frac{m_{seq}}{m}\bar{\epsilon}_{seq}\right) \simeq \Delta\bar{\epsilon}_d - 0.22 \pm 0.22‰$. $\Delta\bar{\epsilon}_d$ can be expanded into contributions from changes in temperature and pCO$_2$:

$$\Delta\bar{\epsilon}_d \simeq \frac{\partial\bar{\epsilon}_d}{\partial T}\Delta\bar{T} + \frac{\partial\bar{\epsilon}_d}{\partial pCO_2}\Delta pCO_2 \quad (14)$$

Given that $\frac{\partial\bar{\epsilon}_d}{\partial T} \simeq -0.105‰$ °C$^{-1}$[34], $\Delta\bar{T} = 2.57 \pm 0.24$°C[48], $\frac{\partial\bar{\epsilon}_d}{\partial pCO_2} \simeq 0.001‰$ ppmv$^{-1}$ (see Section "Dependence of equilibrated air-sea isotopic fractionation on atmospheric CO$_2$" below), and $\Delta pCO_2 \simeq 90$ ppmv, we obtain $\Delta\bar{\epsilon}_d \simeq -0.18 \pm 0.03‰$. Therefore, we conclude that:

$$\Delta\left(\frac{m_{seq}}{m}\bar{\epsilon}_{seq}\right) \simeq -0.40 \pm 0.14‰ \quad (15)$$

Expanding this yields:

$$\left(\bar{\epsilon}_{seq}\frac{m_{seq}}{m}\right)\left(\frac{\Delta m_{seq}}{m_{seq}} - \frac{\Delta m}{m} + \frac{\Delta\bar{\epsilon}_{seq}}{\bar{\epsilon}_{seq}}\right) \simeq -0.40 \pm 0.14‰ \quad (16)$$

We proceed to estimate changes in sequestered carbon, $\Delta m_{seq}$, directly. Rearranging eq. (16) gives:

$$\bar{\epsilon}_{seq}\frac{\Delta m_{seq}}{m} \simeq -0.40 \pm 0.14‰ + \left(\bar{\epsilon}_{seq}\frac{m_{seq}}{m}\right)\left(\frac{\Delta m}{m} - \frac{\Delta\bar{\epsilon}_{seq}}{\bar{\epsilon}_{seq}}\right) \quad (17)$$

Using the same values as above, we obtain for the prefactor: $\left(\bar{\epsilon}_{seq}\frac{m_{seq}}{m}\right) \simeq 40 \times \frac{0.18}{2.30} \simeq 3.1$. We estimate changes in total ocean carbon, $\Delta m$, by summing the estimated $-0.016 \times 10^{18}$ mol carbon transferred from the ocean to the atmosphere[35] with the $-850 \pm 400$ Gt $= -(0.07 \pm 0.03) \times 10^{18}$ mol carbon transferred from the ocean to the terrestrial biosphere[17]. We find that $\Delta m = -(0.09 \pm 0.03) \times 10^{18}$ mol where the negative sign is consistent with carbon leaving the ocean after the LGM. With $m \simeq 3 \times 10^{18}$ mol for the Holocene[50], we find the ratio $\frac{\Delta m}{m} \simeq -0.03 \pm 0.01$.

Now, we estimate changes in the sequestered isotopic offset factor $\bar{\epsilon}_{seq}$. As discussed earlier, decreased air-sea gas exchange would have decreased $\bar{\epsilon}_{seq}$ at the LGM. Furthermore, the lower pCO$_2$ at the LGM would have increased the equilibration time of $\delta^{13}$C, which would have decreased $\bar{\epsilon}_{seq}$ further. Finally, $\bar{\epsilon}_{seq}$ increases with the ratio of the equilibration times of $\delta^{13}$C and DIC. This ratio is proportional to the Revelle buffer factor $B = \frac{\partial \ln pCO_2}{\partial C_{w,sat}}$[33]. Previously, we showed that $B \simeq \frac{C_{w,sat}}{\frac{[CO_3^{2-}]}{O} + [CO_2]} \simeq O\frac{C_{w,sat}}{[CO_3^{2-}]}$ (with $O = -\frac{\partial \ln pCO_2}{\partial[CO_3^{2-}]} \simeq 1.4$)[76]. The buffer factor $O$ is approximately constant and variations in $C_{w,sat}$ are relatively minor, whereas $[CO_3^{2-}]$ is close to inversely proportional to pCO$_2$. Therefore, $B$ would have been lower at the LGM than during the Holocene, which in turn implies a smaller ratio of the equilibration times of $\delta^{13}$C and DIC. Overall, it appears likely that $\bar{\epsilon}_{seq}$ was lower at the LGM than during the Holocene. Therefore, we consider our Holocene estimate $\bar{\epsilon}_{seq} \approx 40‰$ to be an upper bound for the LGM. We also know that $\bar{\epsilon}_{seq}$ must have been

at least as high as photosynthetic fractionation, $\bar{\epsilon}_p = 25‰$[30], at the LGM. We thus obtain a range in sequestered isotopic offset changes, $\Delta\bar{\epsilon}_{seq} \simeq 0$–15‰, and a ratio estimate of $\frac{\Delta\bar{\epsilon}_{seq}}{\bar{\epsilon}_{seq}} \simeq 0.2 \pm 0.2$. Together, we find that $\left(\bar{\epsilon}_{seq}\frac{m_{seq}}{m}\right)\left(\frac{\Delta m}{m} - \frac{\Delta\bar{\epsilon}_{seq}}{\bar{\epsilon}_{seq}}\right) \simeq -0.7 \pm 0.6‰$ and thus:

$$\bar{\epsilon}_{seq}\frac{\Delta m_{seq}}{m} \simeq -1.1 \pm 0.6‰ \quad (18)$$

Using $m \simeq 3 \times 10^{18}$ mol and $\bar{\epsilon}_{seq} \simeq 40‰$, we find:

$$\Delta m_{seq} \simeq (-0.08 \pm 0.05) \times 10^{18} \text{ mol} \quad (19)$$

which amounts to a decrease of $1000 \pm 600$ Pg in sequestered carbon for the Holocene compared to the LGM. In terms of the average deep-ocean sequestered carbon concentration $\bar{C}_{seq}$, this is a change of $\Delta\bar{C}_{seq} = \frac{\Delta m_{seq}}{V} \simeq -0.06 \pm 0.03$ mol/m$^3$ where we have used for the ocean volume $V = 1.4 \times 10^{18}$ m$^3$.

We use the approach laid out in previous publications[68,77] to estimate the increase in atmospheric CO$_2$ induced by such a decrease in the sequestered carbon in the ocean. We begin by writing down the balance equation for carbon in the ocean-atmosphere system:

$$M pCO_2 + V\left(\bar{C}_{sat} + \bar{C}_{seq}\right) = I_{oa} \quad (20)$$

with $M$ the CO$_2$ content of the atmosphere (mol), $V$ the volume of the ocean (m$^3$), and $I_{oa}$ the total carbon inventory of the ocean-atmosphere system (mol).

First, we consider the pCO$_2$ change before the ocean ALK ($A$) budget has re-equilibrated through the carbonate compensation process[32]. In that case, the total carbon in the ocean-atmosphere system is conserved: $\Delta I_{oa} = 0$. Considering changes in the different carbon reservoirs, eq. (20) gives:

$$M\Delta pCO_2 + V\left(\Delta\bar{C}_{sat} + \Delta\bar{C}_{seq}\right) = 0 \quad (21)$$

Expanding $\bar{C}_{sat}$ in terms of pCO$_2$ and rearranging leads to:

$$\Delta pCO_2 \simeq -\frac{1}{\frac{M}{V} + \frac{C_{sat}}{B \times pCO_2}}\Delta\bar{C}_{seq} \quad (22)$$

with $B \equiv \frac{\partial \ln pCO_2}{\partial \ln C_{sat}}$ the Revelle buffer factor[32]. Using $M = 1.8 \times 10^{20}$ mol, $V = 1.4 \times 10^{18}$ m$^3$, $C_{sat} = 2.3$ mol m$^{-3}$, pCO$_2 = 2.5 \times 10^{-4}$ (250 ppmv), and $B = 12$, equation (22) reduces to:

$$\Delta pCO_2 \simeq -1.1 \times 10^{-3}\Delta\bar{C}_{seq} \quad (23)$$

With $\Delta\bar{C}_{seq} \simeq -0.06 \pm 0.03$ mol/m$^3$, eq. (23), we then get the change in pCO$_2$ before carbonate compensation:

$$\Delta pCO_2 \simeq 65 \pm 35 \text{ ppmv} \quad (24)$$

The calculation becomes somewhat more complicated when changes in the total carbon inventory of the ocean-atmosphere system associated with carbonate compensation through dissolution and burial of CaCO$_3$ are considered. In this case, $\Delta I_{oa} = \frac{V\Delta\bar{A}}{2}$, because for every additional mole of carbon in CaCO$_3$, the ocean gains two moles of ALK. To a good approximation, ALK can be approximated by the carbonate alkalinity, i.e., $A \simeq [HCO_3^-] + 2[CO_3^{2-}]$. Neglecting dissolved CO$_2$, $C_{sat} + C_{seq} \simeq [HCO_3^-] + [CO_3^{2-}]$ and $\Delta(C_{sat} + C_{seq}) \simeq \Delta[HCO_3^-] +$

$\Delta[\text{CO}_3^{2-}]$. Thus, $\Delta\bar{A} \approx \Delta\left(\bar{C}_{sat} + \bar{C}_{seq}\right) + \Delta\overline{[\text{CO}_3^{2-}]}$. After full carbonate compensation, $\Delta\overline{[\text{CO}_3^{2-}]}=0$, which means that

$$\Delta\bar{A} \approx \Delta\left(\bar{C}_{sat} + \bar{C}_{seq}\right) \qquad (25)$$

Furthermore, the ocean-atmosphere carbon budget can be written as:

$$M\Delta\text{pCO}_2 + \frac{V\Delta\left(C_{sat} + C_{seq}\right)}{2} = 0. \qquad (26)$$

The factor $\frac{1}{2}$ reflects the very nature of the carbonate compensation process: for every $CO_2$ molecule that the ocean takes up, the ocean also needs to take up a $\text{CO}_3^{2-}$ ion from the sediment to restore the original $\text{CO}_3^{2-}$ concentration. Changes in sequestered carbon lead to changes in ALK but not temperature and salinity. We therefore expand in terms of pCO$_2$ and $A$:

$$\Delta\bar{C}_{sat} = \frac{\bar{C}_{sat}}{B \times \text{pCO}_2}\Delta\text{pCO}_2 + \gamma_A\Delta\bar{A} \qquad (27)$$

with $\gamma_A \equiv \frac{\partial C_{sat}}{\partial A} \simeq 0.90$[78]. Substituting the relationships (27) and (26) in eq. (25), we can solve for $\Delta\,$pCO$_2$:

$$\Delta\text{pCO}_2 \simeq -\frac{1}{2(1-\gamma_A)\frac{M}{V} + \frac{C_{sat}}{B \times \text{pCO}_2}}\Delta\bar{C}_{seq} \qquad (28)$$

Thus,

$$\Delta\text{pCO}_2 \simeq -1.3 \times 10^{-3}\Delta C_{seq} \qquad (29)$$

Again using $\Delta C_{seq} \simeq -0.06 \pm 0.03\,\text{mol/m}^3$, eq. (29) gives:

$$\Delta\text{pCO}_2 \simeq 75 \pm 40\,\text{ppmv} \qquad (30)$$

## Dependence of equilibrated air-sea isotopic fractionation on atmospheric CO$_2$

With increasing atmospheric pCO$_2$, the carbonate equilibrium shifts toward higher $\text{HCO}_3^-$ and lower $\text{CO}_3^{2-}$ concentrations. To estimate the impact of this chemical shift on the air-sea carbon isotope fractionation, we divide the average $\delta^{13}\text{C}$ of the saturated carbon ($\bar{\delta}_{o,sat}$) into contributions from $\text{HCO}_3^-$ and $\text{CO}_3^{2-}$:

$$\bar{\delta}_{o,sat} \simeq \frac{m_{HCO3}}{m_{sat}}\bar{\delta}_{HCO3} + \frac{m_{CO3}}{m_{sat}}\bar{\delta}_{CO3} \qquad (31)$$

with $m_{HCO3}$ and $m_{CO3}$ the saturated $\text{HCO}_3^-$ and $\text{CO}_3^{2-}$ inventories and $\bar{\delta}_{HCO3}$ and $\bar{\delta}_{CO3}$ the average $\delta^{13}\text{C}$ of the saturated $\text{HCO}_3^-$ and $\text{CO}_3^{2-}$. Using $m_{sat} \simeq m_{HCO3} + m_{CO3}$, we can rewrite eq. (31) as:

$$\bar{\delta}_{o,sat} \simeq \bar{\delta}_{HCO3} - \frac{m_{CO3}}{m_{sat}}\bar{\epsilon}_{HCO3,CO3} \qquad (32)$$

with $\epsilon_{HCO3,CO3} = \bar{\delta}_{HCO3} - \bar{\delta}_{CO3}$ the average isotopic fractionation between $\text{HCO}_3^-$ and $\text{CO}_3^{2-}$. This in turn provides an expression for $\epsilon_d$ in terms of contributions from $\text{HCO}_3^-$ and $\text{CO}_3^{2-}$:

$$\epsilon_d = \bar{\delta}_{o,sat} - \bar{\delta}_a \simeq \bar{\epsilon}_{HCO3,a} - \frac{m_{CO3}}{m_{sat}}\bar{\epsilon}_{HCO3,CO3} \qquad (33)$$

with $\epsilon_{HCO3,a} = \bar{\delta}_{HCO3} - \bar{\delta}_a$ the average isotopic fractionation between the saturated $\text{HCO}_3^-$ and the atmosphere. In this expression, the only quantity with a strong dependence on pCO$_2$ is $m_{CO3}$. Therefore, we can approximate:

$$\frac{\partial\epsilon_d}{\partial\text{pCO}_2} \simeq -\frac{1}{m_{sat}}\frac{\partial m_{CO3}}{\partial\text{pCO}_2}\bar{\epsilon}_{HCO3,CO3} \qquad (34)$$

which leads to:

$$\frac{\partial\epsilon_d}{\partial\text{pCO}_2} \simeq \frac{m_{CO3}}{m_{sat}}\frac{\bar{\epsilon}_{HCO3,CO3}}{O \times \text{pCO}_2} \qquad (35)$$

with $O \equiv -\frac{\partial\ln\text{pCO}_2}{\partial[\text{CO}_3^{2-}]} \approx 1.4$[76]. Using $\frac{m_{CO3}}{m_{sat}} = 0.1$, $\bar{\epsilon}_{HCO3,CO3} = 3.5\text{‰}$[34], pCO$_2$ = 250 ppmv, we finally obtain $\frac{\partial\epsilon_d}{\partial\text{pCO}_2} \simeq 0.001\text{‰}\,\text{ppmv}^{-1}$.

## Regenerated carbon budget

For this calculation, we divide the total carbon inventory into preformed and regenerated components. The preformed carbon is defined as the sum of the saturated and disequilibrium carbon. Isotopic bookkeeping analogous to eqs. (7) through (13) then gives:

$$\Delta\bar{\delta}_o = \Delta\bar{\delta}_{o,pre} + \Delta\left(\frac{m_{reg}}{m}\bar{\epsilon}_p\right) \qquad (36)$$

where we used that $\bar{\epsilon}_p = \bar{\delta}_{o,reg} - \bar{\delta}_{o,pre}$ by definition. Eq. (36) can be rearranged as:

$$\Delta m_{reg} = \frac{m}{\bar{\epsilon}_p}\left(\Delta\bar{\delta}_o - \Delta\bar{\delta}_{o,pre}\right) + m_{reg}\frac{\Delta m}{m} \qquad (37)$$

where we have neglected a potential Holocene-LGM difference in the average photosynthetic fractionation factor ($\Delta\bar{\epsilon}_p$), since such effects are thought to be small[30]. To estimate $\Delta m_{reg}$, we use $m = 3 \times 10^{18}\,\text{mol}$, $\bar{\epsilon}_p = -25\text{‰}$[30], $\Delta\bar{\delta}_o = 0.32 \pm 0.10\text{‰}$[46], $\Delta m = -(0.09 \pm 0.03) \times 10^{18}$ mol, i.e., the same values as in the previous calculations. Furthermore, we estimate $\Delta\bar{\delta}_{o,pre}$ based on measured Holocene and LGM $\delta^{13}\text{C}$ in deep-water mass formation regions, as outlined below.

The surface Southern Ocean has low $\delta^{13}\text{C}$ values due to upwelling of water rich in respired carbon. Much of this isotopically light signature is retained at subduction. As a result, deep waters formed in the Southern Ocean have lower $\delta_{o,pre}$ than deep waters sourced from the North Atlantic. Since these deep waters mix conservatively in the ocean interior, we divide $\Delta\bar{\delta}_{o,pre}$ into Northern- and Southern-sourced end-members:

$$\begin{aligned}\Delta\bar{\delta}_{o,pre} &= \bar{\delta}_{o,pre}^{Holo} - \bar{\delta}_{o,pre}^{LGM} \\ &= \bar{\delta}_{o,pre,N}^{Holo}\alpha^{Holo} + \bar{\delta}_{o,pre,S}^{Holo}(1-\alpha^{Holo}) - \left(\bar{\delta}_{o,pre,N}^{LGM}\alpha^{LGM} + \bar{\delta}_{o,pre,S}^{LGM}(1-\alpha^{LGM})\right) \\ &= \left(\Delta\bar{\delta}_{o,pre,N} - \Delta\bar{\delta}_{o,pre,S}\right)\alpha^{Holo} + \Delta\bar{\delta}_{o,pre,S} + \left(\bar{\delta}_{o,pre,N}^{LGM} - \bar{\delta}_{o,pre,S}^{LGM}\right)\Delta\alpha\end{aligned} \qquad (38)$$

with $\alpha^{Holo}$ and $\alpha^{LGM}$ the fractions of deep water from Northern sources during the Holocene and at the LGM. To estimate the preformed $\delta^{13}\text{C}$ of Northern-sourced water for the LGM and Holocene ($\bar{\delta}_{o,pre,N}^{Holo}$ and $\bar{\delta}_{o,pre,N}^{LGM}$), we use published benthic foraminifera $\delta^{13}\text{C}$ from North Atlantic cores located at depths <2000 m. We use benthic foraminifera $\delta^{13}\text{C}$ from Southern Ocean cores located at depths <2000 m to estimate the preformed $\delta^{13}\text{C}$ of Southern-sourced water for the LGM and Holocene ($\bar{\delta}_{o,pre,S}^{Holo}$ and $\bar{\delta}_{o,pre,S}^{LGM}$). We use the cores in which both LGM and Holocene $\delta^{13}\text{C}$ have been measured to estimate the Holocene-LGM difference in preformed $\delta^{13}\text{C}$ of Northern- and Southern-sourced waters ($\Delta\bar{\delta}_{o,pre,N}$ and $\Delta\bar{\delta}_{o,pre,S}$). From these data compilations (see online Supplementary Material), we find $\bar{\delta}_{o,pre,N}^{Holo} = 1.12 \pm 0.06\text{‰}$, $\bar{\delta}_{o,pre,S}^{Holo} = 0.9 \pm 0.1\text{‰}$, $\bar{\delta}_{o,pre,N}^{LGM} = 1.42 \pm 0.05\text{‰}$,

$\bar{\delta}_{o,pre,S}^{LGM} = 0.6 \pm 0.2‰$, $\quad \Delta\bar{\delta}_{o,pre,N} = -0.30 \pm 0.05‰$, $\quad \Delta\bar{\delta}_{o,pre,S} = 0.28 \pm 0.16‰$. Furthermore, we estimate for the Holocene Northern-sourced water-mass fraction $\alpha^{Holo} = 0.40 \pm 0.05$ based on ref. [56]. Now, we use these numbers and eqs. (37) and (38) to create Fig. 3.

Finally, we estimate the global average Holocene-LGM difference in disequilibrium oxygen ($\Delta\bar{O}_{2,dis}$) under the assumption that air-sea gas exchange in the Southern Ocean was completely inhibited at the LGM. Under this scenario, $\Delta\bar{O}_{2,dis} \simeq \frac{\Delta\bar{C}_{dis}}{R_{C:O_2}} = \frac{\Delta m_{dis}}{V R_{C:O_2}}$ (with $\Delta\bar{C}_{dis}$ the Holocene-LGM difference in the global average disequilibrium DIC concentration and $R_{C:O_2}$ the C:O$_2$ Redfield ratio). Furthermore, $\Delta\alpha \simeq 0.3$ under this scenario (see explanation in the main body of the text). Eqs. (37) and (38) then give $\Delta m_{reg} = (-0.005 \pm 0.019) \times 10^{18}$ mol. Thus, $\Delta m_{dis} = \Delta m_{seq} - \Delta m_{reg} = (-0.08 \pm 0.05 + 0.005 \pm 0.019) \times 10^{18}$ mol $= (-0.07 \pm 0.05) \times 10^{18}$ mol. Using $V = 1.4 \times 10^{18}$ m$^3$ and $R_{C:O_2} = 117:170$[79], we obtain: $\Delta\bar{O}_{2,dis} \simeq -0.08 \pm 0.05$ mol/m$^3$.

## Data availability
This study did not generate any new primary data. The compilation of marine $\delta^{13}C$ measurements (secondary data) used to estimate the LGM-to-Holocene change in regenerated carbon is provided in the Supplementary Information/Source Data file (SourceData.xlsx). There are no restrictions on the availability of these data. Source data are provided with this paper.

## Code availability
The Matlab code with which Fig. 3 was generated is available as online Supplementary Material with this article (CalcRegC_d13C_new.m).

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

## Acknowledgements

A.W.O., C.L.F., and J.M.L. are grateful for support from the Simons Collaboration on Computational Biogeochemical Modeling of Marine Ecosystems/CBIOMES (Grant IDs: 549931, MJF; 553242, C.L.F.; 827829, C.L.F.). R.F. is grateful for support through NSF award AGS-1835576. The authors would like to thank Mick Follows, Steph Dutkiewicz, Kat Allen, David McGee, Ed Boyle, and Lorraine Lisiecki for their helpful comments and discussions.

## Author contributions

A.W.O.: conceptualization, analysis, writing initial draft, editing; C.L.F.: conceptualization, analysis, writing initial draft, editing; J.M.L.: analysis, editing; R.F.: conceptualization, analysis, editing.

## Competing interests

The authors declare no competing interests.
