## [Peer Review File · Nature Communications]

Carbon isotope budget indicates biological disequilibrium dominated ocean carbon storage at the LGMREVIEWER COMMENTS

Reviewer #1 (Remarks to the Author):

This is an interesting and novel study that uses isotopic fractionation calculations to estimate a change in the amount of carbon sequestered in the ocean between the LGM and Holocene, in comparison to the observed atmospheric CO₂ change. It follows a methodology similar to a previous study that tracked the ocean's oxygen budget change instead of carbon. The authors point out that the two methods are subject to bias in opposite directions and thus, together, place a useful bracket on the maximum and minimum estimates for ocean carbon sequestration. Overall, this is a very useful exercise with important and interesting results suitable for publication in Nature Communications.

However, the manuscript needs some changes to better explain some of the assumptions made and how they affect the results. In particular, the results have very large uncertainties, whereas the text mostly focuses on the central estimate (not necessarily the most probable) without discussing the implications of the wide uncertainty range. Second, the quantitative results of the study's calculation do not directly address the mechanistic cause of increased carbon sequestration; therefore, the title should not say that isotopes "implicate ice dynamics." Furthermore, some of the explanations presented in the manuscript seem to suffer from circular reasoning, with respect to assuming a priori that sea ice caused a decrease in air-sea gas exchange (e.g., lines 279-281). However, I expect that the authors will be able to resolve these issues with suitable revisions.

Major concerns

1. I am concerned that the manuscript places too much emphasis on comparing the central estimate of their calculated CO₂ change to observations, without enough acknowledgement or discussion of the very wide uncertainty range. The largest source of uncertainty seems to be the amount of carbon transferred from the ocean to the terrestrial biosphere from the LGM to the Holocene, for which this study uses a range of -200 to + 1000 GtC based on one citation. Subsequently, all calculation results using this large range are then provided in the format of center plus/minus half of the range (e.g., 400 +/- 600 GtC). This places too much emphasis on the middle of the uncertainty range, instead of the most probable value. In fact, Figure 5 of the study they cited is a probability density estimate for the change in land storage with a median of 850 GtC with a 1-sigma uncertainty of 400 GtC. Accounting for the change in atmospheric carbon storage and other sources/sinks in the carbon, the "best" estimate C transfer from the ocean to terrestrial storage should still be higher than the central estimate of 400 GtC used in this manuscript and the C transfer is very unlikely to be negative. Thus, the central estimate of their final result is biased by a terrestrial carbon change that is too low. There is also no discussion of the implications of the very large uncertainty range.

A similar issue affects how the study addresses the uncertainty range for the change in mean epsilon_{seq} from the LGM to Holocene. The authors here identify the minimum and maximum possible values for LGM epsilon_{seq} to be 25 to 40 per mil (compared to 40 per mil for the Holocene), and then calculations are performed using an estimated change of 7.5 +/- 7.5 per mil. Thus, the central values of their results inherently emphasize a change in fractionation halfway between its maximum and minimum possible change, without any discussion of whether certain values within that range are more or less likely.

2. Additionally, the manuscript presents a confusing and not entirely convincing explanation of why LGM values of epsilon_{seq} must be smaller than Holocene values. This assumption seems to presuppose that sea ice reduced air-sea gas exchange, and yet this is also the finding that the manuscript purports to demonstrate. Thus, the manuscript appears to employ circular reasoning. To what extent is the very large change in disequilibrium carbon storage (90 +/- 30%) an artifact of assuming that

the LGM ϵ_{seq} was smaller than the Holocene value?

In two places the manuscript attempts to explain the meaning of ϵ_{seq} and how it changes as a function of air-sea gas exchange, but I find the explanations very difficult to follow. The text describes that ϵ_{seq} of a particular water parcel increases as the water spends more time at the surface before subduction; however, this example doesn't help build much intuition about how the whole-ocean ϵ_{seq} might change between the LGM and Holocene when there can be changes in mean ocean $\delta^{13}\text{C}$ and saturated DIC $\delta^{13}\text{C}$ (unlike in the modern day example following a single water mass over a time span of \sim weeks for which those values remain approximately constant). It's also confusing because $m_{\text{w,sat}}$ is defined to be the value when the water subducts whereas $m_{\text{w,seq}}$ is described as increasing over time (ie, before subduction).

3. The manuscript estimates an atmospheric CO_2 change of 90 ± 40 ppm from the sequestered carbon change estimate plus the alkalinity feedback and ocean temperature change effects. Therefore, it is fair to say that ocean carbon sequestration change might explain the entirety of the atmospheric pCO_2 change; however, it might also explain only $\sim 56\%$ of pCO_2 change. Are estimated ocean carbon storage changes of either 90 ppm or 50 ppm equally likely? Without more discussion of the probability distribution of values within that range, the manuscript is too simplistic in its comparison to observed pCO_2 change.

4. The claim presented in the title is also not adequately reflective of the results presented in the paper. The phrase "ice dynamics" in the title is overly vague (sea ice or ice sheets?) and the findings of the paper are that ocean carbon storage change was driven by increased air-sea disequilibrium, which could have been caused by sea ice OR circulation change (lines 332-334). Changes in ocean circulation might have been only very indirectly linked to "ice dynamics."

Minor concerns

5. Typo on line 88: "thes"

6. Line 252: The wording "the sequestered fractionation of the sequestered component of carbon" is hard to understand and sounds awkward.

7. Lines 255-258: Consider switching "higher" and "lower" here to give the change from the LGM to the Holocene (as used in the equations) instead of describing the LGM relative to the Holocene.

8. Line 258: The text should give some indication that this number is derived in the methods section.

9. Section 5.2: Here you use distinguish "preformed" and "regenerated" DIC components whereas the rest of the manuscript uses "regenerated" and "disequilibrium" as categories for DIC components. Consider a brief explanation reconciling the difference in terminology here.

10. Line 488: I recommend larger error bars for the fractions of LGM northern and southern water masses allowing for possibly more northern source water. (The lower bound for northern sourced water is probably fine.) There almost certainly was some mixing of northern and southern water along their boundary in the Atlantic, and so the fraction of northern water in the deep Pacific could have been non-negligible (20% or more) as suggested by higher $\delta^{13}\text{C}$ values in the deep Pacific than the deep South Atlantic.

Reviewer #2 (Remarks to the Author):

Omta and coauthors provide a novel assessment of glacial-interglacial changes in the ocean biological pump, based on an analysis of published foraminifera $\delta^{13}\text{C}$ measurements. Their result agrees well with the oxygen-based estimate of Vollmer et al., which - because of the difference of air-sea exchange dynamics between the gases - is taken to imply a very strong impediment to air-sea exchange was exerted by sea ice during the glacial. I think this is an interesting and thought-provoking paper that makes a valuable contribution, and is worth publishing. My comments are mostly in regards to prior work and methodological details.

The received wisdom in ocean biogeochemistry has tended to view carbon as being prone to disequilibrium effects, while oxygen - with its much quicker equilibration timescale - is relatively immune to disequilibrium. Some have shown this is not strictly true in the modern ocean, but it has nonetheless coloured a lot of thinking about how to interpret paleo records of O_2 vs. $\delta^{13}\text{C}$ and D_{14}C . The current work suggests that glacial-interglacial changes in O_2 disequilibrium may have been similar to glacial-interglacial changes in DIC, which is interesting. I feel this is a bit understated in the current manuscript, and could be better highlighted. Furthermore, although I don't disagree, I don't feel the paper makes a clear case why the results 'implicate ice dynamics' - if this is to be in the title, it ought to be better explained.

In this vein, the results imply a very large disequilibrium change for O_2 . It would be interesting to see some quantitative discussion of the implied amounts, and whether or not these O_2 disequilibria - in terms of mmol m^{-3} - appear plausible. I note that Eggleston and Galbraith (Climate of the Past, 2018) show model simulations that may be helpful in this regard (their section 3.5, Fig 8).

There is a lot to think about in the methodology, which was novel to me, and made some interesting assumptions. I did not have the opportunity to think as thoroughly about it as I would have liked, but I did not spot any logical errors, and it seemed reasonable as far as I could tell. That said, it was not clear to me that the "pCO₂ effect" discussed by Galbraith et al. (GBC 2015) was taken into account here - this acts by increasing the exchange timescale during the glacial (not just changing the distribution of isotopes through speciation) and it seems to me that it is therefore not included in ϵ_{d} . I'm not sure how the effect would propagate through the terms defined by the authors, but it may contribute to the overall disequilibrium.

The authors should also discuss how their results relate to those of Moree et al (Climate of the Past, 2021) and Khatiwala et al. (Science Advances, 2019) both of whom undertake related exercises.

Substituting the terms, as given, into eq. 3 does not obviously yield eq. 4. Please rephrase with clearer sign conventions.

It seems the authors should either use Gebbie 2015 for mean LGM $\delta^{13}\text{C}$, or say why this is not used.

Reviewer #3 (Remarks to the Author):

In this study, the authors seek to account for the amplitude of glacial-interglacial atmospheric CO₂ change using a novel and elegant approach that targets stable carbon isotopes.

I have reviewed an earlier version of this study elsewhere, and the present version is greatly improved on that previous version. The version that I have seen before was already a very elegant study, but was arguably hampered by its focus on respired carbon alone (disequilibrium and gas-exchange were cast aside). Here, the authors make a big step forward in expanding their approach to also consider

disequilibrium carbon and the role of gas-exchange.

It is remarkable that this shift of focus has resulted in a “180 degree” change in the study’s conclusions, from the claim that “the full glacial-interglacial CO₂ shift is due to changes in the biological export of carbon out of the surface ocean”, to the current claim that “the dominant effect [on CO₂ sequestration in the glacial ocean] appears to have been a stronger air-sea carbon disequilibrium in the Southern Ocean at the LGM”. I find this to be a great improvement: it resonates with numerous previous radiocarbon-based studies (still not cited in this study, sadly), but it also goes further than these previous studies in proposing to constrain the relative magnitudes of the disequilibrium carbon versus respired carbon contributions. The latter is really new and important.

Overall, I find the study’s conclusions to be well supported, particularly regarding its qualitative claims. The quantitative estimates provided by the study are highly uncertain, but they are nicely bounded by limiting scenarios, and they are precisely what is needed to move debates regarding LGM CO₂ sequestration beyond the well-established status quo (a status quo that is clearly enunciated in the paper’s introduction).

This an engaging and useful contribution to the scientific literature and should be published in my view; however, I do think that this should be subject to some minor but important corrections. I list these below:

1. Referencing: in my view the study lacks some important referencing, both regarding the use of stable carbon isotope gradients to infer carbon cycle implications, and the use of carbon isotopes or oxygen etc. to provide tentative quantitative estimates of the combined CO₂ sequestration effects of gas-exchange and the biological pump. For starters, it is hard to ignore that stable carbon isotope gradient approaches are rooted in the pioneering work of e.g. Broecker 1982 and Shackleton 1983. It also seems an omission to ignore the efforts of e.g. Peterson and Lisiecki (2018) amongst others. The very early studies were far removed from what is now possible (and what is achieved in the present study), and the more recent efforts have lacked any quantitative analyses, so it only elevates the present work to position it relative to these prior efforts based on stable carbon isotopes.

Further, I also think it is a major omission to leave out reference to other studies that have sought to provide quantitative estimates of the CO₂ sequestration effects of gas-exchange and ocean ‘ventilation’ (i.e. biopump efficiency). I apologise for citing my own work, but at least in doing so I am 100% confident about the content and proposals made in the studies: e.g. Skinner et al. (EPSL, 2015) used deep Pacific radiocarbon to estimate the sequestered carbon impact to be ~49 ppm; Gottschalk et al. (Nat Comms, 2016), used oxygen estimates and radiocarbon in the deep Southern Ocean to infer that at least half of the glacial-interglacial CO₂ change could be accounted for potentially; Skinner et al. (Nat Comms, 2017) used a global radiocarbon database to estimate the CO₂ impact at ~65ppm; and more recently Skinner et al. (2023) used a revised global radiocarbon database to update this estimate to ~50+/-27ppm. There are probably other estimates out there. My main point is that the manuscript shouldn’t ignore previous studies such as these, as they are based on complementary approaches and only strengthen the case made by the authors.

2. Terminology: the authors should be applauded for shifting away from what might be called the ‘Princeton consensus’, where everything is driven by biological export productivity and nutrients, towards a view that incorporates the auxiliary effects of residence times, and gas-exchange especially. However, throughout the manuscript, and in the title especially, the authors insist on referring to ‘biologically sequestered’ carbon, when in fact this is just ‘sequestered carbon’ (I submit). Yes, biology cycles nearly ALL carbon in the ocean eventually, but this doesn’t mean that it always remains in the ocean instead of the atmosphere because of biology. Water upwelling in the Southern Ocean, may

have elevated DIC largely due to respired carbon being added to it at some point in the past (biology is a proximal cause of its elevated DIC), but the ultimate cause of the carbon staying in the water at the sea surface is disequilibrium arising from gas-exchange and upper ocean mixing. It is surely best to remain agnostic in general therefore, and refer to 'sequestered carbon' simply.

Below I include a some more detailed remarks on the text:

Abstract and title: despite the clear conclusions of the study, which make biological export a relatively minor contributor to glacial-interglacial CO₂ change as compared to disequilibrium, the authors appear to downplay the role of gas-exchange in the title and abstract. I would strongly suggest to rectify this: it will reflect the findings of the study more accurately and will be noticed, hopefully by those who have ignored disequilibrium for too long.

Line 46: it is a bit of a stretch to describe this is an 'emerging consensus' when it has been pretty clear since at least 2008 (Brovkin et al., 2008 etc.), and is now the main subject of review papers. How about 'long-standing consensus'?

Line 66: I suggest to add the existing estimates of how much the deglacial terrestrial carbon storage change would have affected atmospheric CO₂ (~-18ppm?).

Line 75: if we include the terrestrial carbon contribution we are basically at square one again.

Line 93: I suggest to change this to "...the amount of carbon in the abyssal ocean is larger than can be accounted for by respired carbon alone". I really do not think we can refer to it as "biological carbon" without causing a great deal of confusion. Notably, we don't refer to the fraction of respired carbon that made it into the atmosphere as 'biological carbon in the atmosphere"! I also suggest that the authors state here that this 'excess carbon' is referred to as 'disequilibrium carbon'. I would suggest to further describe how this disequilibrium carbon can be positive or negative (i.e. a deficit of carbon uptake, as in the North Atlantic), and can further be divided into carbon anomalies that stem from biological or physical pathways but always arise from limited gas-exchange efficiency (i.e. a physical process).

Line 113: I really think this should be corrected to "a change in the biological and solubility pumps", or "a change in the biological pump and air-sea gas exchange", or (perhaps optimally) "a change in ocean-atmosphere carbon partitioning". It ceases to be 'biological carbon' when it has had the chance to become 'equilibrium carbon' but missed it.

Figure 1. This figure could be improved perhaps, to show atmospheric CO₂ for context, and the marine stable carbon isotope data that are used, for example.

Line 175: a reference is needed here maybe (e.g. Williams & Follows, 2011; Eggleston & Galbraith, 2018)?

Line 223: I find it confusing to think of this term as a 'fractionation' factor, when it is referring to an isotopic *offset* that arises for a mix of processes. Would it not be better to refer to it as an 'isotopic offset' or similar, especially given definition on line 236?

Line 248: Should this not be: "Therefore... gas-exchange, all else being equal"?

Line 273: I think it would be better to refer to glacial-interglacial cycles, rather than Ice Ages (which are sometimes confused with 'Ice House' states, such as the entire Quaternary).

Line 314: “.. between *an* oxygen-based estimate..” (there are others). I also think it is worth noting that the very same holds true for e.g. a series of radiocarbon-based estimates (e.g. Skinner et al., 2015; Skinner et al., 2017; Skinner and Bard 2022; Skinner et al., 2023), which suggest e.g. 45ppm, 65 ppm, 50 +/- 27 ppm... (again with apologies for self-citation).

Line 327: please remove ‘biologically’; as soon as we include disequilibrium carbon effects we are just referring to ‘sequestered carbon’ (again, there is no complement of this ‘biological carbon’ in the sky).

Line 337: as above, please remove ‘biologically’.

Line 341: as above, please remove biological.

Line 343: as above.

Line 351: Because of my own ‘research baggage’ I can’t help but note that Skinner et al. (2019), as well as Skinner and Bard (2022) and Skinner et al. (2023) have all pointed to the apparent effects of air-sea gas exchange inefficiency (on radiocarbon, and carbon sequestration), particularly in the intermediate depth ocean. Skinner et al. (2020, 2013, 2015) also made a case for gas-exchange in the Southern Ocean, again based on radiocarbon (combined with Nd isotopes etc.). These results all resonate with the present study’s findings. I apologise again for citing my own studies (especially as there are surely others having made the same point), and I am not suggesting that they should all be cited, but I have worked on precisely this issue for many years now and I feel it is important to underline that the conclusion of limited Southern Ocean gas-exchange (or indeed just ‘normal’ Southern Ocean gas exchange; Skinner 2008) contributing to CO₂ sequestration at the LGM is not that new. Noting this ‘consilience’ doesn’t diminish the present study at all, as it arrives at the same conclusion in a new and elegant way.

Line 377 to 393: much of this text is verbatim from the main manuscript (e.g. line 234). I suggest to remove any unnecessary repetition here.

Line 396: What is the reference for this mean ocean temperature at the LGM? Perhaps use the MOT change from e.g. Bereiter et al. (2018), ~-2.5oC? Does this make any difference?

Line 487: Although reference 51 is a very interesting study, I don’t think it can be cited as demonstrating volumetric changes in Southern sourced water at the LGM – data studies would be better for that (e.g. Lund et al., 2011). (Again, I can’t help but note here that Skinner (2008) made the case for this volumetric change in southern sourced water having a direct impact on atmospheric CO₂ precisely via the disequilibrium carbon that southern sourced water retains, suggesting an effect of ~27ppm just from the volumetric effect and modern gas-exchange efficiency alone).

Supplementary material: The references given for the data are for data compilations, not the original data. Please update to include references for the original data. Also, please include a statement of what benthic foraminifer species have been used in each case, and if any (and what magnitude) ‘corrections’ may have been applied, e.g. to bring the measured values onto a “Uvigerina sp. consistent” scale for example.

I hope that the authors will find my comments constructive and useful, as I sincerely intend them to be.

References (with apologies for all the self-citation!):

- Shackleton, N. J., M. A. Hall, J. Line and C. Shuxi (1983). "Carbon isotope data in core V19-30 confirm reduced carbon dioxide concentration in the ice age atmosphere." *Nature* 306(5941): 319-322.
- Broecker, W. S. (1982). "Glacial to interglacial changes in ocean chemistry." *Progress in Oceanography* 11: 151-197.
- Peterson, C. D. and L. E. Lisiecki (2018). "Deglacial carbon cycle changes observed in a compilation of 127 benthic $\delta^{13}\text{C}$ time series (20–6 ka)." *Clim. Past* 14(8): 1229-1252.
- Brovkin, V., A. Ganopolski, D. Archer and G. Munhoven (2012). "Glacial CO₂ cycle as a succession of key physical and biogeochemical processes." *Climate of the Past* 8: 251-246.
- Williams, R. G. and M. Follows (2011). *Ocean Dynamics and the Carbon Cycle: Principals and Mechanisms*. Cambridge UK, Cambridge University Press.
- Skinner, L. C., F. Primeau, E. Freeman, M. de la Fuente, P. Goodwin, J. Gottschalk, E. Huang, I. N. McCave, T. Noble and A. E. Scriver (2017). "Radiocarbon constraints on the 'glacial' ocean circulation and its impact on atmospheric CO₂." *Nature Communications* 8: 16010.
- Skinner, L., I. N. McCave, L. Carter, S. Fallon, A. Scriver and F. Primeau (2015). "Reduced ventilation and enhanced magnitude of the deep Pacific carbon pool during the last glacial period." *Earth and Planetary Science Letters* 411: 45-52.
- Skinner, L., F. Muschitiello and A. E. Scriver (2019). "Marine Reservoir Age Variability Over the Last Deglaciation: Implications for Marine Carbon Cycling and Prospects for Regional Radiocarbon Calibrations." *Paleoceanography and Paleoclimatology* 34(11): 1807-1815.
- Skinner, L., F. Primeau, A. Jeltsch-Thömmes, F. Joos, P. Köhler and E. Bard (2023). "Rejuvenating the ocean: mean ocean radiocarbon, CO₂ release, and radiocarbon budget closure across the last deglaciation." *Clim. Past* 19(11): 2177-2202.
- Skinner, L. C. (2009). "Glacial-interglacial atmospheric CO₂ change: a possible standing volume effect on deep ocean carbon sequestration." *Climate of the Past* 5: 537-550.

REVIEWER COMMENTS

Reviewer #1 (Remarks to the Author):

This is an interesting and novel study that uses isotopic fractionation calculations to estimate a change in the amount of carbon sequestered in the ocean between the LGM and Holocene, in comparison to the observed atmospheric CO₂ change. It follows a methodology similar to a previous study that tracked the ocean’s oxygen budget change instead of carbon. The authors point out that the two methods are subject to bias in opposite directions and thus, together, place a useful bracket on the maximum and minimum estimates for ocean carbon sequestration. Overall, this is a very useful exercise with important and interesting results suitable for publication in *Nature Communications*.

We would like to thank Reviewer #1 for this excellent summary of our work and for the encouraging words.

However, the manuscript needs some changes to better explain some of the assumptions made and how they affect the results. In particular, the results have very large uncertainties, whereas the text mostly focuses on the central estimate (not necessarily the most probable) without discussing the implications of the wide uncertainty range. Second, the quantitative results of the study’s calculation do not directly address the mechanistic cause of increased carbon sequestration; therefore, the title should not say that isotopes “implicate ice dynamics.” Furthermore, some of the explanations presented in the manuscript seem to suffer from circular reasoning, with respect to assuming a priori that sea ice caused a decrease in air-sea gas exchange (e.g., lines 279–281). However, I expect that the authors will be able to resolve these issues with suitable revisions.

We would like to thank Reviewer #1 for their comments, which have helped improve the manuscript considerably. Below, we give a point-by-point response to each of the concerns.

Major concerns

1. I am concerned that the manuscript places too much emphasis on comparing the central estimate of their calculated CO₂ change to observations, without enough acknowledgement or discussion of the very wide uncertainty range. The largest source of uncertainty seems to be the amount of carbon transferred from the ocean to the terrestrial biosphere from the LGM to the Holocene, for which this study uses a range of -200 to +1000 Gt C based on one citation. Subsequently, all calculation results using this large range are then provided in the format of center plus/minus half of the range (e.g., 400±600 Gt C). This places too much emphasis on the middle of the uncertainty range, instead of the most probable value. In fact, Figure 5 of the study they cited is a probability density estimate for the change in land storage with a median of 850 Gt C with a 1- σ uncertainty of 400 Gt C. Accounting for the change in atmospheric carbon storage and other sources/sinks in the carbon, the “best” estimate C transfer from the ocean to terrestrial storage should still be higher than the central estimate of 400 Gt C used in this manuscript and the C transfer is very unlikely to be negative. Thus, the central estimate of their final result is biased by a terrestrial carbon change that is too low. There is also no discussion of the implications of the very large uncertainty range.

We appreciate the Reviewer’s concern over how we discussed the error ranges in our estimates and have done our best to deal directly with this issue for the different parts of our calculation (see later responses). For the specific case of terrestrial carbon mentioned here, we agree that the median of the Jeltsch-Thömmes et al. (2019) estimate of the Holocene-LGM difference in terrestrial carbon storage (Δm_{land}) is larger than its center value and apologize for this oversight. We have gone ahead and redone the calculations (see Methods section 5.1) using $\Delta m_{land} = 850 \pm 400$ Gt C, as suggested by the Reviewer. Fortunately, this does not have a significant impact on our results, since Δm_{land} represents only a minor contribution (part of the second term $\frac{\Delta m}{m}$ in eqs. 5 and 17) in our calculation of Δm_{seq} .

This is because after full equilibration, Δm_{land} mainly affects the average $\delta^{13}C$ of the ocean-atmosphere system rather than the $\delta^{13}C$ difference between the ocean and atmosphere that our calculations are based on. We appreciate the opportunity to clarify this point in the main text as it is a critical one. In section 2 of the main text, we have added the following sentences to make it clear how our data analysis strategy helps us remove the effects of processes like terrestrial carbon emission: “As the absolute changes in the size of other mobile carbon stores (such as terrestrial biomass) are small relative to the entire ocean-atmosphere carbon inventory, the effect of these reservoirs on the total CO_2 change after equilibration must also be small. By focusing on changes in the overall difference between atmospheric and deep ocean $\delta^{13}C$ during periods of slow change (the LGM and the Holocene), we link the vertical gradients in isotopic composition directly to changes in quasi-steady state carbon storage in the sea.”

A similar issue affects how the study addresses the uncertainty range for the change in mean ϵ_{seq} from the LGM to Holocene. The authors here identify the minimum and maximum possible values for LGM ϵ_{seq} to be 25 to 40‰ (compared to 40‰ for the Holocene), and then calculations are performed using an estimated change of 7.5 ± 7.5 ‰. Thus, the central values of their results inherently emphasize a change in fractionation halfway between its maximum and minimum possible change, without any discussion of whether certain values within that range are more or less likely.

This is in fact a major source of uncertainty in our calculation. We don't have any information about the probability distribution for $\Delta \bar{\epsilon}_{seq}$ within the 0–15‰ range, other than that the extreme values appear unlikely (see further explanation under point 3 below). We have included the following explanation in Section 3.2 of the revised manuscript:

“As we don't have any further information about the probability distribution of $\Delta \bar{\epsilon}_{seq}$, we estimate $\Delta \bar{\epsilon}_{seq} = 7.5 \pm 7.5$ ‰ (with the central values more likely than the extreme ones).”

2. Additionally, the manuscript presents a confusing and not entirely convincing explanation of why LGM values of ϵ_{seq} must be smaller than Holocene values. This assumption seems to pre-suppose that sea ice reduced air-sea gas exchange, and yet this is also the finding that the manuscript purports to demonstrate. Thus, the manuscript appears to employ circular reasoning. To what extent is the very large change in disequilibrium carbon storage ($90 \pm 30\%$) an artifact of assuming that the LGM ϵ_{seq} was smaller than the Holocene value? For our calculation of the change in sequestered carbon (Δm_{seq}), we needed an estimate of $\bar{\epsilon}_{seq}$ at the LGM. In making this estimate, we considered which LGM to Holocene changes would have impacted $\bar{\epsilon}_{seq}$. These changes (of which the increase in air-sea gas exchange is one) suggest that $\bar{\epsilon}_{seq}$ was lower at the LGM than during the Holocene. Subsequently, we made a calculation of the change in regenerated carbon (Δm_{reg}), for which we did not need to make any assumptions about air-sea gas exchange. If we would have found $\Delta m_{reg} \gtrsim \Delta m_{seq}$, then that would have suggested that reduced air-sea gas exchange at the LGM played no major role in the enhanced ocean carbon sequestration. The Δm_{reg} estimate is rather sensitive to the assumed LGM water-mass configuration (see also reply to point 10 below and the new Fig. 3). If the water-mass configuration didn't change from the LGM to the Holocene, then our analysis is inconclusive as to whether $\Delta m_{reg} < \Delta m_{seq}$. However, if we assume a significant shoaling of the Northern overturning cell at the LGM, then we find Δm_{reg} to be significantly smaller than Δm_{seq} . The latter regime provides a more self-consistent picture that is also consistent with $\Delta^{14}C$ and oxygen proxies. This is internally consistent with reduced air-sea gas exchange at the LGM, although we agree that this is not a definitive proof. In addition to including the new Figure 3, we have rewritten essentially all of Section 3.3 and much of Section 4 to explain and discuss this.

In two places the manuscript attempts to explain the meaning of ϵ_{seq} and how it changes as a function of air-sea gas exchange, but I find the explanations very difficult to follow. The text describes that ϵ_{seq} of a particular water parcel increases as the water spends more time at the surface before subduction; however, this example doesn't help build much intuition about how the whole-ocean ϵ_{seq} might change between the LGM and Holocene when there can be changes in mean ocean $\delta^{13}C$ and saturated DIC $\delta^{13}C$ (unlike in the modern day example following a single water mass over a time span of \sim weeks for which those values

remain approximately constant).

It is unlikely that the differences in mean ocean and saturated $\delta^{13}\text{C}$ between the LGM and the Holocene significantly affected $\bar{\epsilon}_{\text{seq}}$. That said, $\bar{\epsilon}_{\text{seq}}$ could have been affected by the Holocene-LGM difference in $p\text{CO}_2$ due to associated changes in equilibration times. Nevertheless, the upper and lower boundary for $\bar{\epsilon}_{\text{seq}}$ remain 40 and 25‰. We have included a discussion of these effects in Section 3.2 of the revised manuscript:

“The Holocene-LGM difference in the sequestered isotopic offset ($\Delta\bar{\epsilon}_{\text{seq}}$) can be estimated by realizing that $\bar{\epsilon}_{\text{seq}}$ is always larger than $\sim 25\text{‰}$ (the photosynthetic fractionation) and that the LGM value of $\bar{\epsilon}_{\text{seq}}$ must have been lower than the Holocene value of $\sim 40\text{‰}$ for two main reasons. Firstly, the expanded sea-ice cover at the LGM would have led to an increase in the disequilibrium carbon. Secondly, $\bar{\epsilon}_{\text{seq}}$ increases with the ratio of the equilibration times of $\delta^{13}\text{C}$ and DIC. This ratio was likely larger at the LGM than during the Holocene, as explained in the Methods section.”

with the following elaboration under Methods (Section 5.1):

“Finally, $\bar{\epsilon}_{\text{seq}}$ increases with the ratio of the equilibration times of $\delta^{13}\text{C}$ and DIC. This ratio is proportional to the Revelle buffer factor $B = \frac{\partial \ln p\text{CO}_2}{\partial \ln C_{w,\text{sat}}}$ (Galbraith et al., 2015). Previously, we showed that $B \simeq \frac{C_{w,\text{sat}}}{\frac{[\text{CO}_3^{2-}]}{O} + [\text{CO}_2]} \simeq O \frac{C_{w,\text{sat}}}{[\text{CO}_3^{2-}]}$ (with $O = -\frac{\partial \ln p\text{CO}_2}{\partial \ln [\text{CO}_3^{2-}]} \simeq 1.4$, Omta et al., 2010).

The buffer factor O is approximately constant and variations in $C_{w,\text{sat}}$ are relatively minor, whereas $[\text{CO}_3^{2-}]$ is close to inversely proportional to $p\text{CO}_2$. Therefore, B would have been lower at the LGM than during the Holocene, which in turn implies a smaller ratio of the equilibration times of $\delta^{13}\text{C}$ and DIC.”

It’s also confusing because $m_{w,\text{sat}}$ is defined to be the value when the water subducts whereas $m_{w,\text{seq}}$ is described as increasing over time (ie, before subduction).

We agree that the definition of $m_{w,\text{seq}}$ was confusing and inconsistent with the definition of, e.g., m_{sat} ; our apologies for that. For clarification, we have renamed $m_{w,\text{seq}}$ and m_w as $C_{w,\text{seq}}$ and $C_{w,\text{sat}}$, respectively. Furthermore, we have included definitions of δ_w , $\delta_{w,\text{sat}}$, $C_{w,\text{seq}}$, $C_{w,\text{sat}}$, and $\epsilon_{w,\text{seq}}$ in Section 3.2 of the revised manuscript:

“Here, δ_w is the $\delta^{13}\text{C}$ value of the water mass, $\delta_{w,\text{sat}}$ is the saturated $\delta^{13}\text{C}$ at the surface of the Southern Ocean, $C_{w,\text{seq}}$ is the biologically sequestered carbon in the water mass, $C_{w,\text{sat}}$ is the saturated DIC concentration at the surface of the Southern Ocean, and $\epsilon_{w,\text{seq}}$ is the $\delta^{13}\text{C}$ difference between the saturated and sequestered DIC in the water mass.”

3. The manuscript estimates an atmospheric CO_2 change of 90 ± 40 ppm from the sequestered carbon change estimate plus the alkalinity feedback and ocean temperature change effects. Therefore, it is fair to say that ocean carbon sequestration change might explain the entirety of the atmospheric $p\text{CO}_2$ change; however, it might also explain only $\sim 56\%$ of $p\text{CO}_2$ change. Are estimated ocean carbon storage changes of either 90 ppm or 50 ppm equally likely? Without more discussion of the probability distribution of values within that range, the manuscript is too simplistic in its comparison to observed $p\text{CO}_2$ change.

We don’t know the probability distribution of the sequestered carbon change. What we can say is that the extreme values of our estimated range represent extreme scenarios that appear rather unlikely. One end of the range is represented by $\Delta\bar{\epsilon}_{\text{seq}} = 0\text{‰}$, which gives a $p\text{CO}_2$ change of only 34 ppmv. This scenario seems unlikely, because the Holocene-LGM difference in atmospheric $p\text{CO}_2$ alone would have led to $\Delta\bar{\epsilon}_{\text{seq}} > 0\text{‰}$ (see explanation under point 2 above). To compensate for this and get to $\Delta\bar{\epsilon}_{\text{seq}} = 0\text{‰}$, air-sea gas exchange would have had to be enhanced at the LGM compared to the Holocene. The other end of the range is represented by $\Delta\bar{\epsilon}_{\text{seq}} = 15\text{‰}$, which gives a $p\text{CO}_2$ change of 115 ppmv. This scenario doesn’t seem likely either, since it would have required the complete absence of gas exchange in the Southern Ocean at the LGM. We have included this explanation in Section 3.2 of the revised manuscript:

“Thus, we have lower and upper bounds of 0 and 15‰ for $\Delta\bar{\epsilon}_{\text{seq}}$, which represent unlikely extreme scenarios. $\Delta\bar{\epsilon}_{\text{seq}} = 0\text{‰}$ requires air-sea gas exchange to be larger at the LGM, when taking into account that $p\text{CO}_2$ was lower at LGM than during the Holocene. $\Delta\bar{\epsilon}_{\text{seq}} = 15$ requires the absence of any gas exchange in the Southern Ocean at the LGM.”

4. The claim presented in the title is also not adequately reflective of the results presented in the paper. The phrase “ice dynamics” in the title is overly vague (sea ice or ice sheets?) and the findings of the paper are that ocean carbon storage change was driven by increased air-sea disequilibrium, which could have been caused by sea ice OR circulation change (lines 332–334). Changes in ocean circulation might have been only very indirectly linked to “ice dynamics.”

This is a valid point. We have changed the manuscript title to “Isotopes Constrain Mechanisms of Carbon Storage at the Last Glacial Maximum”.

Minor concerns

5. Typo on line 88: “thes”

Good catch! We have replaced “thes” with “these”.

6. Line 252: The wording “the sequestered fractionation of the sequestered component of carbon” is hard to understand and sounds awkward.

Yes, that wording was confusing and awkward. We have reformulated this sentence as: “The LGM-to-Holocene change in the ocean-atmosphere $\delta^{13}\text{C}$ difference can now be divided into contributions from physical air-sea fractionation and from the sequestered carbon pool”.

7. Lines 255–258: Consider switching “higher” and “lower” here to give the change from the LGM to the Holocene (as used in the equations) instead of describing the LGM relative to the Holocene.

We have reformulated this sentence as:

“Taking the LGM-to-Holocene change in average ocean temperature equal to $2.57 \pm 0.24^\circ\text{C}$ (Bereiter et al., 2018) and the $p\text{CO}_2$ change equal to 90 ppmv (Bereiter et al., 2015), we estimate $\Delta\bar{\epsilon}_d = -0.18 \pm 0.03\text{‰}$ ”.

8. Line 258: The text should give some indication that this number is derived in the methods section.

We have added “(see Methods section)” here.

9. Section 5.2: Here you use distinguish “preformed” and “regenerated” DIC components whereas the rest of the manuscript uses “regenerated” and “disequilibrium” as categories for DIC components. Consider a brief explanation reconciling the difference in terminology here.

Great point! To introduce this change in terminology, we have added the following text at the beginning of what is now Section 5.3:

“For this calculation, we divide the total carbon inventory into preformed and regenerated components. The preformed carbon is defined as the sum of the saturated and disequilibrium carbon. Isotopic bookkeeping analogous to eqs. (7) through (13) then gives”.

10. Line 488: I recommend larger error bars for the fractions of LGM northern and southern water masses allowing for possibly more northern source water. (The lower bound for northern sourced water is probably fine.) There almost certainly was some mixing of northern and southern water along their boundary in the Atlantic, and so the fraction of northern water in the deep Pacific could have been non-negligible (20% or more) as suggested by higher $\delta^{13}\text{C}$ values in the deep Pacific than the deep South Atlantic.

We agree that there could have been non-negligible mixing of Northern-sourced waters into the deep Pacific. Furthermore, there exists significant uncertainty about how much North Atlantic Deep Water shoaled at the LGM. Generally, studies based on $\delta^{13}\text{C}$ indicate more shoaling than those based on ϵ_{Nd} (Pavia et al., 2022). To address this uncertainty in the LGM water-mass configuration, we have created the new Figure 3 depicting $\frac{\Delta m_{\text{reg}}}{\Delta m_{\text{seq}}}$ as a function of the change in the Northern-sourced water-mass fraction ($\Delta\alpha$). For $\Delta\alpha = 0$ (no change), the regenerated component accounts for between -20% and +110% of the sequestered carbon change, with a most likely value of +50%. In other words, our analysis is inconclusive about the contribution of regenerated carbon to the change in sequestered carbon in this regime. For $\Delta\alpha = 0.3$ (Northern-sourced water being confined to the upper

2000 m of the Atlantic at the LGM), the regenerated component accounts for between -40% and +50% of the sequestered carbon change, with a most likely value of +7%. In other words, regenerated carbon accounts for no more than half of the change in sequestered carbon in this regime. As we mentioned in our reply under point 2, the latter regime provides a more self-consistent picture that is also consistent with $\Delta^{14}\text{C}$ and oxygen proxies. In addition to including the new Figure 3, we have rewritten essentially all of Section 3.3 and much of Section 4 to explain and discuss this.

References:

- Bereiter, B., S. Eggleston, J. Schmitt, C. Nehrbaß-Ahles, T.F. Stocker, H. Fischer, S. Kipfstuhl and J. Chappellaz (2015), Geophysical Research Letters 42: 542–549*
- Bereiter, B., S. Shackleton, D. Baggenstos, K. Kawamura and J. Severinghaus (2018), Nature 553: 39–44*
- Galbraith, E.D., E.Y. Kwon, D. Bianchi, M.P. Hain and J.L. Sarmiento (2015), Global Biogeochemical Cycles 29: 307–324*
- Jeltsch-Thömmes, A., G. Battaglia, O. Cartapanis, S.L. Jaccard and F. Joos (2019), Climate of the Past 15: 849–879*
- Omta, A.W., P. Goodwin and M.J. Follows (2010), Global Biogeochemical Cycles 24: GB3008*
- Pavia, F.J., C.S. Jones and S.K. Hines (2022), Journal of Climate 35: 5465–5482*

Reviewer #2 (Remarks to the Author):

Omta and coauthors provide a novel assessment of glacial-interglacial changes in the ocean biological pump, based on an analysis of published foraminifera $\delta^{13}\text{C}$ measurements. Their result agrees well with the oxygen-based estimate of Vollmer et al., which – because of the difference of air-sea exchange dynamics between the gases – is taken to imply a very strong impediment to air-sea exchange was exerted by sea ice during the glacial. I think this is an interesting and thought-provoking paper that makes a valuable contribution, and is worth publishing. My comments are mostly in regards to prior work and methodological details.

We would like to thank Reviewer #2 for these kind words and for the stimulating and helpful comments, which have helped us improve the manuscript considerably. Below, we address the comments point-by-point.

The received wisdom in ocean biogeochemistry has tended to view carbon as being prone to disequilibrium effects, while oxygen – with its much quicker equilibration timescale – is relatively immune to disequilibrium. Some have shown this is not strictly true in the modern ocean, but it has nonetheless coloured a lot of thinking about how to interpret paleo records of O_2 vs. $\delta^{13}\text{C}$ and $\Delta^{14}\text{C}$. The current work suggests that glacial-interglacial changes in O_2 disequilibrium may have been similar to glacial-interglacial changes in DIC, which is interesting. I feel this is a bit understated in the current manuscript, and could be better highlighted.

We appreciate the reviewers interest in this point from our manuscript and have added additional text and a calculation to address this point (see response further down).

Furthermore, although I don't disagree, I don't feel the paper makes a clear case why the results 'implicate ice dynamics' – if this is to be in the title, it ought to be better explained.

We agree that the implications about ice dynamics are rather indirect. Therefore, we have changed the manuscript title to "Isotopes Constrain Mechanisms of Carbon Storage at the Last Glacial Maximum".

In this vein, the results imply a very large disequilibrium change for O_2 . It would be interesting to see some quantitative discussion of the implied amounts, and whether or not these O_2 disequilibria – in terms of mmol m^{-3} – appear plausible. I note that Eggleston and Galbraith (Climate of the Past, 2018) show model simulations that may be helpful in this regard (their section 3.5, Fig 8).

The implied global average O_2 disequilibrium change would be $80 \pm 50 \mu\text{M}$, broadly consistent with the difference between the "Moderate World" and "Cold World" simulations of Eggleston & Galbraith (2018). We have included a brief discussion of the O_2 disequilibrium in Section 3.4:

"We estimate (see Methods section) that this would imply a Holocene-LGM difference in global ocean average disequilibrium oxygen $\Delta\bar{\text{O}}_{2,\text{dis}} \simeq -80 \pm 50 \mu\text{M}$. This is broadly consistent with the results from a recent modeling experiment (see Figs. 8b and 8c in Eggleston & Galbraith, 2018)."

The calculation is in Section 5.3:

"Finally, we estimate the global average Holocene-LGM difference in disequilibrium oxygen ($\Delta\bar{\text{O}}_{2,\text{dis}}$) under the assumption that air-sea gas exchange in the Southern Ocean was completely inhibited at the LGM. Under this scenario, $\Delta\bar{\text{O}}_{2,\text{dis}} \simeq \frac{\Delta\bar{\text{C}}_{\text{dis}}}{R_{\text{C}:\text{O}_2}} = \frac{\Delta m_{\text{dis}}}{V R_{\text{C}:\text{O}_2}}$ (with $\Delta\bar{\text{C}}_{\text{dis}}$ the Holocene-LGM difference in the global average disequilibrium DIC concentration and $R_{\text{C}:\text{O}_2}$ the C: O_2 Redfield ratio). Furthermore, $\Delta\alpha \simeq 0.3$ under this scenario (see explanation in the main body of the text). Eqs. (37) and (38) then give $\Delta m_{\text{reg}} = (-0.005 \pm 0.019) \times 10^{18} \text{ mol}$. Thus, $\Delta m_{\text{dis}} = \Delta m_{\text{seq}} - \Delta m_{\text{reg}} = (-0.08 \pm 0.05 + 0.005 \pm 0.019) \times 10^{18} \text{ mol} = (-0.07 \pm 0.05) \times 10^{18} \text{ mol}$. Using $V = 1.4 \times 10^{18} \text{ m}^3$ and $R_{\text{C}:\text{O}_2} = 117:170$ (Lenton & Watson, 2000), we obtain: $\Delta\bar{\text{O}}_{2,\text{dis}} \simeq -0.08 \pm 0.05 \text{ mol/m}^3$."

There is a lot to think about in the methodology, which was novel to me, and made some interesting assumptions. I did not have the opportunity to think as thoroughly about it as I would have liked, but I did not spot any logical errors, and it seemed reasonable as

far as I could tell. That said, it was not clear to me that the “pCO₂ effect” discussed by Galbraith et al. (GBC 2015) was taken into account here – this acts by increasing the exchange timescale during the glacial (not just changing the distribution of isotopes through speciation) and it seems to me that it is therefore not included in ϵ_d . I’m not sure how the effect would propagate through the terms defined by the authors, but it may contribute to the overall disequilibrium.

Great point! $\bar{\epsilon}_{seq}$ increases with the ratio of the equilibration times of $\delta^{13}C$ and DIC. This ratio is proportional to the Revelle buffer factor $B \equiv \frac{\partial \ln pCO_2}{\partial \ln C_{w,sat}}$, which was likely lower at the LGM. We have included a discussion of these effects in Section 3.2 of the revised manuscript:

“The Holocene-LGM difference in the sequestered isotopic offset ($\Delta \bar{\epsilon}_{seq}$) can be estimated by realizing that $\bar{\epsilon}_{seq}$ is always larger than $\sim 25\text{‰}$ (the photosynthetic fractionation) and that the LGM value of $\bar{\epsilon}_{seq}$ must have been lower than the Holocene value of $\sim 40\text{‰}$ for two main reasons. Firstly, the expanded sea-ice cover at the LGM would have led to an increase in the disequilibrium carbon. Secondly, $\bar{\epsilon}_{seq}$ increases with the ratio of the equilibration times of $\delta^{13}C$ and DIC. This ratio was likely larger at the LGM than during the Holocene, as explained in the Methods section.”

with the following elaboration under Methods (Section 5.1):

“Finally, $\bar{\epsilon}_{seq}$ increases with the ratio of the equilibration times of $\delta^{13}C$ and DIC. This ratio is proportional to the Revelle buffer factor $B = \frac{\partial \ln pCO_2}{\partial \ln C_{w,sat}}$ (Galbraith et al., 2015). Previously, we showed that $B \simeq \frac{C_{w,sat}}{\frac{[CO_3^{2-}]}{3} + [CO_2]} \simeq O \frac{C_{w,sat}}{[CO_3^{2-}]}$ (with $O = -\frac{\partial \ln pCO_2}{\partial \ln [CO_3^{2-}]} \simeq 1.4$, Omta et al., 2010).

The buffer factor O is approximately constant and variations in $C_{w,sat}$ are relatively minor, whereas $[CO_3^{2-}]$ is close to inversely proportional to pCO₂. Therefore, B would have been lower at the LGM than during the Holocene, which in turn implies a smaller ratio of the equilibration times of $\delta^{13}C$ and DIC.”

The authors should also discuss how their results relate to those of Morée et al (Climate of the Past, 2021) and Khatiwala et al. (Science Advances, 2019) both of whom undertake related exercises.

We agree that the model sensitivity studies of Khatiwala et al. (2019) and Morée et al. (2021) are complementary to our work. These studies investigate the potential impacts of different processes on CO₂ and/or $\delta^{13}C$ in a model framework. However, it is difficult to assess the relative importance of these processes based on such simulations alone. Our data-based calculations suggest that differences in Southern Ocean disequilibrium DIC played a major role in glacial-interglacial CO₂ changes, which further narrows the range of potential processes. Thus, our work provides constraints for future modeling efforts. We now discuss this complementarity at the end of Section 4 of the revised manuscript:

“The main contribution of our work is to provide data-based constraints on the LGM carbon and $\delta^{13}C$ budgets that have primarily been studied through the lense of numerical simulations (Heinze et al., 1991; Broecker et al., 1999; Lauderdale et al., 2013; Khatiwala et al., 2019; Morée et al., 2018; 2021; Stein et al., 2020). Simulations rely on assumptions in the model formulations, for example with respect to changes in air-sea gas exchange and carbon export. Our data-based calculations suggest that differences in Southern Ocean carbon disequilibrium played a major role in glacial-interglacial CO₂ changes, which narrows down the range of potential processes.”

Substituting the terms, as given, into eq. 3 does not obviously yield eq. 4. Please rephrase with clearer sign conventions.

Yes, this was not obvious, our apologies for the confusion. For clarification, we have rephrased the paragraph between eqs. 3 and 4 and eq. 4 itself:

“We take $\Delta \bar{\delta}_o = 0.32 \pm 0.10\text{‰}$ (1- σ uncertainty) (Gebbie et al., 2015) and $\Delta \bar{\delta}_a = 0.10 \pm 0.10\text{‰}$ (Eggleston et al., 2016). Taking the LGM-to-Holocene change in average ocean temperature equal to $2.57 \pm 0.24\text{°C}$ (Bereiter et al., 2018) and the pCO₂ change equal to 90 ppmv (Bereiter et al., 2015), we estimate $\Delta \bar{\epsilon}_d = -0.18 \pm 0.03\text{‰}$ (see Methods section). Substituting these values in eq. (3), we then estimate: $\Delta \left(\frac{m_{seq}}{m} \bar{\epsilon}_{seq} \right) \simeq \Delta \bar{\epsilon}_d - \Delta \bar{\delta}_o + \Delta \bar{\delta}_a \simeq -0.4 \pm 0.2\text{‰}$.”

It seems the authors should either use Gebbie 2015 for mean LGM $\delta^{13}\text{C}$, or say why this is not used.

We have redone the calculations using the Gebbie et al. (2015) value of $0.32 \pm 0.10\text{‰}$ ($1\text{-}\sigma$ error) for $\Delta\bar{\delta}_o$. This did not have a significant impact on the results.

References:

- Bereiter, B., S. Shackleton, D. Baggenstos, K. Kawamura and J. Severinghaus (2018), *Nature* 553: 39–44
- Bouttes, N., D. Paillard, D.M. Roche and V. Brovkin (2011), *Geophysical Research Letters* 38: L02705
- Broecker, W.S., J. Lynch-Stieglitz, D.E. Archer, M. Hoffmann, E. Maier-Reimer, O. Marchal, T. Stocker and N. Gruber (1999), *Global Biogeochemical Cycles* 13: 817–820
- Eggleston, S., J. Schmitt, B. Bereiter, R. Schneider and H. Fischer (2016), *Paleoceanography* 31: 434–452
- Eggleston, S., and E.D. Galbraith (2018), *Biogeosciences* 15: 3761–3777
- Galbraith, E.D., E.Y. Kwon, D. Bianchi, M.P. Hain and J.L. Sarmiento (2015), *Global Biogeochemical Cycles* 29: 307–324
- Gebbie, G., C.D. Peterson, L.E. Lisiecki and H.J. Spero (2015), *Quaternary Science Reviews* 125: 144–159
- Heinze, C., E. Maier-Reimer and K. Winn (1991), *Paleoceanography* 6: 395–430
- Khatiwala, S., A. Schmittner and J. Muglia (2019), *Science Advances* 5: eaaw4981
- Lauderdale, J.M., A.C. Naveira Garabato, K.I.C. Oliver, M.J. Follows and R.G. Williams (2013), *Climate Dynamics* 41: 2145–2164
- Lenton, T.M. and A.J. Watson (2000), *Global Biogeochemical Cycles* 14: 225–248
- Morée, A.L., J. Schwinger and C. Heinze (2018), *Biogeosciences* 15: 7205–7223
- Morée, A.L., J. Schwinger, U.S. Ninnemann, A. Jeltsch-Thömmes, I. Bethke and C. Heinze (2021), *Climate of the Past* 17: 753–774
- Omta, A.W., P. Goodwin and M.J. Follows (2010), *Global Biogeochemical Cycles* 24: GB3008
- Stein, K., A. Timmermann, E.Y. Kwon and T. Friedrich (2020), *Proceedings of the National Academy of Sciences* 117: 4498–4504

Reviewer #3 (Remarks to the Author):

In this study, the authors seek to account for the amplitude of glacial-interglacial atmospheric CO₂ change using a novel and elegant approach that targets stable carbon isotopes. I have reviewed an earlier version of this study elsewhere, and the present version is greatly improved on that previous version. The version that I have seen before was already a very elegant study, but was arguably hampered by its focus on respired carbon alone (disequilibrium and gas-exchange were cast aside). Here, the authors make a big step forward in expanding their approach to also consider disequilibrium carbon and the role of gas-exchange. It is remarkable that this shift of focus has resulted in a “180 degree” change in the study’s conclusions, from the claim that “the full glacial-interglacial CO₂ shift is due to changes in the biological export of carbon out of the surface ocean”, to the current claim that “the dominant effect [on CO₂ sequestration in the glacial ocean] appears to have been a stronger air-sea carbon disequilibrium in the Southern Ocean at the LGM”. I find this to be a great improvement: it resonates with numerous previous radiocarbon-based studies (still not cited in this study, sadly), but it also goes further than these previous studies in proposing to constrain the relative magnitudes of the disequilibrium carbon versus respired carbon contributions. The latter is really new and important.

Please see our responses to the “Referencing” point below. We have added referencing to prior important work focusing on both carbon isotopes.

Overall, I find the study’s conclusions to be well supported, particularly regarding its qualitative claims. The quantitative estimates provided by the study are highly uncertain, but they are nicely bounded by limiting scenarios, and they are precisely what is needed to move debates regarding LGM CO₂ sequestration beyond the well-established status quo (a status quo that is clearly enunciated in the paper’s introduction).

We would like to thank Reviewer #3 for these encouraging words and for the stimulating and helpful comments, which have helped us improve the manuscript considerably. Below, we address the comments point-by-point.

This an engaging and useful contribution to the scientific literature and should be published in my view; however, I do think that this should be subject to some minor but important corrections. I list these below:

1. Referencing: in my view the study lacks some important referencing, both regarding the use of stable carbon isotope gradients to infer carbon cycle implications, and the use of carbon isotopes or oxygen etc. to provide tentative quantitative estimates of the combined CO₂ sequestration effects of gas-exchange and the biological pump. For starters, it is hard to ignore that stable carbon isotope gradient approaches are rooted in the pioneering work of e.g. Broecker 1982 and Shackleton 1983. It also seems an omission to ignore the efforts of e.g. Peterson and Lisiecki (2018) amongst others. The very early studies were far removed from what is now possible (and what is achieved in the present study), and the more recent efforts have lacked any quantitative analyses, so it only elevates the present work to position it relative to these prior efforts based on stable carbon isotopes.

We agree that these studies provide relevant background to our work, since they establish that the vertical $\delta^{13}\text{C}$ gradient provides a measure of sequestered carbon in the ocean. Therefore, we have included these references at the end of the Introduction:

“We focus on the vertical $\delta^{13}\text{C}$ gradient in the ocean, which provides another measure of sequestered carbon (Broecker, 1982; Shackleton et al., 1983; Broecker & McGee, 2013; Peterson & Lisiecki, 2018).”

Further, I also think it is a major omission to leave out reference to other studies that have sought to provide quantitative estimates of the CO₂ sequestration effects of gas-exchange and ocean ‘ventilation’ (i.e. biopump efficiency). I apologise for citing my own work, but at least in doing so I am 100% confident about the content and proposals made in the studies: e.g. Skinner et al. (EPSL, 2015) used deep Pacific radiocarbon to estimate the sequestered carbon impact to be ~49 ppm; Gottschalk et al. (Nat Comms, 2016), used oxygen estimates and radiocarbon in the deep Southern Ocean to infer that at least half of the glacial-

interglacial CO₂ change could be accounted for potentially; Skinner et al. (Nat Comms, 2017) used a global radiocarbon database to estimate the CO₂ impact at ~65 ppm; and more recently Skinner et al. (2023) used a revised global radiocarbon database to update this estimate to ~50±27 ppm. There are probably other estimates out there. My main point is that the manuscript shouldn't ignore previous studies such as these, as they are based on complementary approaches and only strengthen the case made by the authors.

We agree that the approaches based on radiocarbon and oxygen and our approach based on $\delta^{13}C$ complement each other for two main reasons:

1. It is remarkable how well the different methods agree on the amount of sequestered carbon. We emphasize this at the beginning of Section 3.4 of the revised manuscript: "Our $\delta^{13}C$ -based estimate of the LGM to Holocene change in pCO₂ (65±35 ppmv) is in close agreement with $\Delta^{14}C$ -(50±27 ppmv, Skinner et al., 2023) and oxygen-based estimates (64±28 ppmv, Vollmer et al., 2022)".

We chose to focus on these two studies here, since these are the most recent estimates of glacial-interglacial changes in sequestered carbon.

2. The different approaches all point to disequilibrium DIC as a major contributor to glacial-interglacial CO₂ changes, which indeed strengthens the case. We emphasize this in Section 4 of the revised manuscript:

"In other words, the waters rich in biologically sequestered carbon that upwelled at the surface in the Southern Ocean were much further from equilibrium with the atmosphere at the LGM compared to today. Such an increase in disequilibrium carbon is consistent with an expansion of the sea-ice coverage around Antarctica (Stephens & Keeling, 2000; Skinner et al., 2017; Jansen, 2017) in combination with a smaller Northern overturning cell."

We hope that the additional discussion and references help contextualize our isotopic calculation more firmly within the existing literature.

2. Terminology: the authors should be applauded for shifting away from what might be called the 'Princeton consensus', where everything is driven by biological export productivity and nutrients, towards a view that incorporates the auxiliary effects of residence times, and gas-exchange especially. However, throughout the manuscript, and in the title especially, the authors insist on referring to 'biologically sequestered' carbon, when in fact this is just 'sequestered carbon' (I submit). Yes, biology cycles nearly ALL carbon in the ocean eventually, but this doesn't mean that it always remains in the ocean instead of the atmosphere because of biology. Water upwelling in the Southern Ocean, may have elevated DIC largely due to respired carbon being added to it at some point in the past (biology is a proximal cause of its elevated DIC), but the ultimate cause of the carbon staying in the water at the sea surface is disequilibrium arising from gas-exchange and upper ocean mixing. It is surely best to remain agnostic in general therefore, and refer to 'sequestered carbon' simply.

We agree that not only export productivity but also DIC disequilibrium is important. However, the disequilibrium is largely due to respired carbon, as the Reviewer correctly points out. Therefore, we think that 'biologically sequestered carbon' is an appropriate term to capture both the regenerated and (biological) disequilibrium components of the ocean carbon pool. Furthermore, we are concerned that the term 'sequestered carbon' is insufficiently specific. For example, it could also include carbon storage through entirely abiotic mechanisms such as a change in the average ocean temperature. To avoid confusion, we have taken care to explain what we mean by 'biologically sequestered carbon' and we have provided a rationale for this terminology:

"Since disequilibrium carbon has a primarily biological origin, we combine the regenerated and disequilibrium carbon and refer to their sum as the biologically sequestered carbon (similar to Skinner & Bard, 2022): $m_{seq} = m_{reg} + m_{dis}$ " (end of Section 3.1).

Below I include a some more detailed remarks on the text:

Abstract and title: despite the clear conclusions of the study, which make biological export a relatively minor contributor to glacial-interglacial CO₂ change as compared to disequilibrium, the authors appear to downplay the role of gas-exchange in the title and abstract. I would strongly suggest to rectify this: it will reflect the findings of the study more accurately and will be noticed, hopefully by those who have ignored disequilibrium for too long. *Great point! To emphasize the central role of disequilibrium, we have added the following sentences to the Abstract:*

“An analysis of the carbon isotopic signatures of different water masses indicates similar regenerated carbon inventories at the LGM and during the Holocene. This suggests that the enhanced carbon storage at the LGM is best explained as a result of dampened air-sea gas exchange, rather than increased export productivity.”

Line 46: it is a bit of a stretch to describe this is an ‘emerging consensus’ when it has been pretty clear since at least 2008 (Brovkin et al., 2008 etc.), and is now the main subject of review papers. How about ‘long-standing consensus’?

OK, we have changed “emerging consensus” to “long-standing consensus”. Furthermore, we have included Brovkin et al. (2007; 2012) as a reference here.

Line 66: I suggest to add the existing estimates of how much the deglacial terrestrial carbon storage change would have affected atmospheric CO₂ (~18 ppm?).

We have included the ~-20 ppmv estimate by Davies-Barnard et al. (2017) here.

Line 75: if we include the terrestrial carbon contribution we are basically at square one again.

We have changed this sentence to:

“Thus, accounting for the changes in terrestrial biomass, ocean temperature, and salinity leaves essentially the entire ~90 ppmv of the atmospheric CO₂ rise unexplained.”

Line 93: I suggest to change this to “...the amount of carbon in the abyssal ocean is larger than can be accounted for by respired carbon alone”. I really do not think we can refer to it as “biological carbon” without causing a great deal of confusion. Notably, we don’t refer to the fraction of respired carbon that made it into the atmosphere as ‘biological carbon in the atmosphere’! I also suggest that the authors state here that this ‘excess carbon’ is referred to as ‘disequilibrium carbon’.

We have changed this sentence to:

“This “disequilibrium DIC” increases the amount of carbon stored in the ocean, in addition to carbon storage due to sinking of organic matter (Toggweiler et al., 2003; Lauderdale et al., 2013; Galbraith & Skinner, 2020).”

I would suggest to further describe how this disequilibrium carbon can be positive or negative (i.e. a deficit of carbon uptake, as in the North Atlantic), and can further be divided into carbon anomalies that stem from biological or physical pathways but always arise from limited gas-exchange efficiency (i.e. a physical process).

This is a great suggestion. We have included a discussion of disequilibrium carbon and its physical and biological origins here:

“Disequilibrium DIC can also have a physical origin as is the case in the North Atlantic. Here the cooling of Northward-flowing surface water results in a positive disequilibrium, i.e. the DIC of the surface waters is lower than the saturated value. This has been termed “physical disequilibrium DIC” as opposed to the “biological disequilibrium DIC” (Khatiwala et al., 2019) that we will be focusing on.”

Line 113: I really think this should be corrected to “a change in the biological and solubility pumps”, or “a change in the biological pump and air-sea gas exchange”, or (perhaps optimally) “a change in ocean-atmosphere carbon partitioning”. It ceases to be ‘biological carbon’ when it has had the chance to become ‘equilibrium carbon’ but missed it.

In line with our definition (see our reply under point 2), we have changed this to “a change in biologically sequestered carbon in the ocean”.

Figure 1. This figure could be improved perhaps, to show atmospheric CO₂ for context, and the marine stable carbon isotope data that are used, for example.

After taking a careful look at Figure 1, we agreed that it could be improved. We have added a new panel with atmospheric pCO₂ from 20 until 6 kyr BP, which has become Fig. 1a. We have tried to make the δ¹³C plot (now Fig. 1b) clearer by increasing the vertical distance between atmospheric and oceanic δ¹³C and adding arrows indicating the larger ocean-atmosphere δ¹³C difference during the Holocene than at the LGM. The marine δ¹³C data used for this figure is the Peterson & Lisiecki (2018) compilation, which also included a map of the core site locations (Fig. 1 in Peterson & Lisiecki, 2018).

Line 175: a reference is needed here maybe (e.g. Williams & Follows, 2011; Eggleston & Galbraith, 2018)?

We have included these references here.

Line 223: I find it confusing to think of this term as a ‘fractionation’ factor, when it is referring to an isotopic *offset* that arises for a mix of processes. Would it not be better to refer to it as an ‘isotopic offset’ or similar, especially given definition on line 236?

That is a great suggestion, we have changed the terminology accordingly.

Line 248: Should this not be: “Therefore... gas-exchange, all else being equal”?

That is correct, we have changed this.

Line 273: I think it would be better to refer to glacial-interglacial cycles, rather than Ice Ages (which are sometimes confused with ‘Ice House’ states, such as the entire Quaternary). Good point, we have replaced “Ice Ages” with “glacial-interglacial cycles”.

Line 314: “...between *an* oxygen-based estimate...” (there are others). I also think it is worth noting that the very same holds true for e.g. a series of radiocarbon-based estimates (e.g. Skinner et al., 2015; Skinner et al., 2017; Skinner and Bard 2022; Skinner et al., 2023), which suggest e.g. 45 ppm, 65 ppm, 50±27 ppm... (again with apologies for self-citation). *We have rewritten this paragraph to be a comparison with both the Vollmer et al. (2022) oxygen-based estimate and the Skinner et al. (2023) Δ¹⁴C-based estimate (see reply to point 1 above). We chose to focus on these two studies in particular, since these are the most recent estimates of glacial-interglacial changes in sequestered carbon.*

Line 327: please remove ‘biologically’; as soon as we include disequilibrium carbon effects we are just referring to ‘sequestered carbon’ (again, there is no complement of this ‘biological carbon’ in the sky).

Line 337: as above, please remove ‘biologically’.

Line 341: as above, please remove biological.

Line 343: as above.

This is essentially the discussion about the ‘biologically sequestered carbon’ terminology (point 2). Again, we would like to emphasize that the disequilibrium is largely due to respired carbon. Therefore, we think that ‘biologically sequestered carbon’ is an appropriate term to capture both the regenerated and (biological) disequilibrium components of the ocean carbon pool.

Line 351: Because of my own ‘research baggage’ I can’t help but note that Skinner et al. (2019), as well as Skinner and Bard (2022) and Skinner et al. (2023) have all pointed to the apparent effects of air-sea gas exchange inefficiency (on radiocarbon, and carbon sequestration), particularly in the intermediate depth ocean. Skinner et al. (2020, 2013, 2015) also made a case for gas-exchange in the Southern Ocean, again based on radiocarbon (combined with Nd isotopes etc.). These results all resonate with the present study’s findings. I apologise again for citing my own studies (especially as there are surely others having made the same point), and I am not suggesting that they should all be cited, but I have worked on precisely this issue for many years now and I feel it is important to underline that the conclusion of limited Southern Ocean gas-exchange (or indeed just ‘normal’ Southern Ocean gas exchange; Skinner 2008) contributing to CO₂ sequestration at the LGM is not that new. Noting this ‘consilience’ doesn’t diminish the present study

at all, as it arrives at the same conclusion in a new and elegant way.

This is a valid point. We have included citations to earlier studies that reached similar conclusions here, in particular Gottschalk et al. (2016), Skinner et al. (2017; 2019; 2021), and Pavia et al (2022).

Line 377 to 393: much of this text is verbatim from the main manuscript (e.g. line 234). I suggest to remove any unnecessary repetition here.

We have removed almost this entire paragraph, because it was indeed mostly a verbatim repetition of text from Section 3.2 of the main manuscript.

Line 396: What is the reference for this mean ocean temperature at the LGM? Perhaps use the MOT change from e.g. Bereiter et al. (2018), $\sim -2.5^{\circ}\text{C}$? Does this make any difference? *We have redone the calculations using the Bereiter et al. (2018) estimate $\Delta T = 2.57 \pm 0.24^{\circ}\text{C}$. This did not have a significant impact on the final results of the calculations.*

Line 487: Although reference 51 is a very interesting study, I don't think it can be cited as demonstrating volumetric changes in Southern sourced water at the LGM – data studies would be better for that (e.g. Lund et al., 2011). (Again, I can't help but note here that Skinner (2008) made the case for this volumetric change in southern sourced water having a direct impact on atmospheric CO_2 precisely via the disequilibrium carbon that southern sourced water retains, suggesting an effect of ~ 27 ppm just from the volumetric effect and modern gas-exchange efficiency alone).

We have replaced this reference with Lund et al. (2011).

Supplementary material: The references given for the data are for data compilations, not the original data. Please update to include references for the original data. Also, please include a statement of what benthic foraminifer species have been used in each case, and if any (and what magnitude) 'corrections' may have been applied, e.g. to bring the measured values onto a "Uvigerina sp. consistent" scale for example.

In the revised Supplementary Material spreadsheet, we have included a column with the references to the original data and a column with the benthic foraminifera species used in each case. None of the original source articles mentioned that any corrections were applied to the raw $\delta^{13}\text{C}$ measurements. The Uvigerina equivalent scale with associated corrections is used for $\delta^{18}\text{O}$ rather than for $\delta^{13}\text{C}$ (Thornalley et al., 2010).

I hope that the authors will find my comments constructive and useful, as I sincerely intend them to be.

We would like to thank the Reviewer once again for the comments, which we did find constructive and useful.

References:

- Bereiter, B., S. Shackleton, D. Baggenstos, K. Kawamura and J. Severinghaus (2018), *Nature* 553: 39–44
- Broecker, W.S. (1982), *Progress in Oceanography* 11: 151–197
- Broecker, W.S., and D. McGee (2013), *Earth and Planetary Science Letters* 368: 175–182
- Brovkin, V., A. Ganopolski, D.E. Archer and S. Rahmstorf (2007), *Paleoceanography* 22: PA4202
- Brovkin, V., A. Ganopolski, D.E. Archer and G. Munhoven (2012), *Climate of the Past* 8: 251–246
- Davies-Barnard, T., A. Ridgwell, J. Singarayer and P. Valdes (2017), *Climate of the Past* 13: 1381–1401
- Eggleston, S., and E.D. Galbraith (2018), *Biogeosciences* 15: 3761–3777
- Galbraith, E.D., and L.C. Skinner (2020), *Annual Review of Marine Science* 12: 559–586
- Gottschalk, J., L.C. Skinner, J. Lippold, H. Vogel, N. Frank, S.L. Jaccard and C. Waelbroeck (2016), *Nature Communications* 7: 11539
- Lauderdale, J.M., A.C. Naveira Garabato, K.I.C. Oliver, M.J. Follows and R.G. Williams (2013), *Climate Dynamics* 41: 2145–2164
- Lund, D.C., J.F. Adkins and R. Ferrari (2011), *Paleoceanography and Paleoclimatology* 26: PA1213
- Pavia, F.J., C.S. Jones and S.K. Hines (2022), *Journal of Climate* 35: 5465–5482
- Peterson, C.D., and L.E. Lisiecki (2018), *Climate of the Past* 14: 1229–1252
- Shackleton, N.J., M.A. Hall, J. Line and C. Shuxi (1983), *Nature* 306: 319–322
- Skinner, L.C. (2009), *Climate of the Past* 5: 537–550
- Skinner, L.C., A.E. Scrivner, D. Vance, S. Barker, S. Fallon and C. Waelbroeck (2013), *Geology* 41: 667–670
- Skinner, L.C., I.N. McCave, L. Carter, S. Fallon, A.E. Scrivner and F. Primeau (2015), *Earth and Planetary Science Letters* 411: 45–52
- Skinner, L.C., F. Primeau, E. Freeman, M. de la Fuente, P. Goodwin, J. Gottschalk, E. Huang, I.N. McCave, T. Noble and A.E. Scrivner (2017), *Nature Communications* 8: 16010
- Skinner, L.C., F. Muschitiello and A.E. Scrivner (2019), *Paleoceanography and Paleoclimatology* 34: 1807–1815
- Skinner, L.C., E. Freeman, D. Hodell, C. Waelbroeck, N. Vazquez Riveiros and A.E. Scrivner (2021), *Paleoceanography and Paleoclimatology* 36: e2020PA004074
- Skinner, L.C. and E. Bard (2022), *Reviews of Geophysics* 60: e2020RG000720
- Skinner, L.C., F. Primeau, A. Jeltsch-Thömmes, F. Joos, P. Köhler and E. Bard (2023), *Climate of the Past* 19: 2177–2202
- Thornalley, D.J.R., H. Elderfield and I.N. McCave (2010), *Paleoceanography* 25: PA1211
- Toggweiler, J.R., R. Murnane, S. Carson, A. Gnanadesikan and J.L. Sarmiento (2003), *Global Biogeochemical Cycles* 17: 1027
- Vollmer, T.D., T. Ito and J. Lynch-Stieglitz (2022), *Paleoceanography and Paleoclimatology* 37: e2021PA004339
- Williams, R.G., and M.J. Follows (2011), *Ocean Dynamics and the Carbon Cycle: Principles and Mechanisms*. Cambridge UK, Cambridge University Press.

REVIEWERS' COMMENTS

Reviewer #1 (Remarks to the Author):

The revised manuscript is much improved, and I now recommend it for publication. However, I advise one minor change, which is to acknowledge that the estimate of 75 +/- 40 ppm associated with biological sequestration may not explain the full 90 ppm of observed CO₂ change. This is important especially because the estimates based on oxygen and d13C budgets are in such extremely good agreement. At least three times the manuscript points out that the biological sequestration estimate (75 +/-40) "is sufficient to explain the entire CO₂ shift" (lines 28, 312, 406) without ever acknowledging the 15 ppm difference between the two numbers. I suggest adding a sentence near the end of the manuscript that discusses potential mechanisms to explain the remaining 15 ppm difference.

-Lorraine Lisiecki

Reviewer #1 (Remarks on code availability):

The code runs and produces results that match what is shown in Figure 3.

Reviewer #2 (Remarks to the Author):

I was reviewer #2 for the initial submission. I think the authors have done a good job of responding to my comments, and have no further follow-up on these. I only have a few small remaining suggestions regarding the title and abstract.

I think the new title is too general - the authors only use stable carbon isotopes, and 'constrain mechanisms' is vague and leaves out the main point. I would suggest something like, "Global carbon isotope assessment indicates biological disequilibrium dominated ocean carbon storage at the LGM".

Abstract

I don't feel this is accurate: 'Here we show that stable carbon isotopes can be used to identify the driving mechanism.'

The isotopes do not allow identification of a single driving mechanism, they only help to characterize the carbon storage. Furthermore, this makes it sound like nobody used stable carbon isotopes for this problem before, when in fact they've been used for many decades. The novelty lies in the details of the calculation and size of the dataset. Please rephrase this.

I also think the abstract should mention the word disequilibrium somewhere. Perhaps something like: An analysis of the carbon isotopic signatures of different water masses indicates similar regenerated carbon inventories at the LGM and during the Holocene, REQUIRING THAT THE CHANGE IN STORAGE WAS DOMINATED BY DISEQUILIBRIUM.

Reviewer #3 (Remarks to the Author):

The authors have greatly improved their manuscript, in my opinion, based on all the reviewers' input, and I have no further comments, apart from one. It seems the authors will not budge on this one, so I don't wish to make a big deal of it; however, I still do not agree that carbon retained by the ocean due to limited air-sea gas exchange can be referred to as 'biologically sequestered carbon'. I think it is

clear that biology did not do the 'sequestering' in this case (even if it did the 'pumping'). If air-sea exchange was made more efficient (i.e. there was no sea ice, or the ocean circulation was altered etc.), but there was NO change in biological activity, the sequestration could be diminished or removed (i.e. abiotically). That said, the point about avoiding conflation with solubility effects is a valid one, and I agree that my suggestions were therefore not perfect. I just feel that the chosen terminology is unfortunate in directing attention away from the physical processes that are actually doing the 'sequestering' and that are receiving renewed attention in this study - these are physical processes that will need to be studied further and therefore might benefit from being highlighted. In any event, I look forward to seeing the study in print, and discussing it further in other forums!

REVIEWERS' COMMENTS

Reviewer #1 (Remarks to the Author):

The revised manuscript is much improved, and I now recommend it for publication.

We would like to thank Reviewer #1 Lorraine Lisiecki for her thoughtful comments in both the previous and current Review rounds, which have helped improve the manuscript considerably.

However, I advise one minor change, which is to acknowledge that the estimate of 75 ± 40 ppm associated with biological sequestration may not explain the full 90 ppm of observed CO_2 change. This is important especially because the estimates based on oxygen and $\delta^{13}\text{C}$ budgets are in such extremely good agreement. At least three times the manuscript points out that the biological sequestration estimate (75 ± 40) “is sufficient to explain the entire CO_2 shift” (lines 28, 312, 406) without ever acknowledging the 15 ppm difference between the two numbers.

Although 90 ppm lies within the margin of error of our estimation, we agree that biological sequestration may not account for the full 90 ppm of observed CO_2 change. Therefore, we have softened these statements from “is sufficient” to “could be sufficient”.

I suggest adding a sentence near the end of the manuscript that discusses potential mechanisms to explain the remaining 15 ppm difference.

Given the excellent correspondence between the estimates based on O_2 and $\delta^{13}\text{C}$, any remaining unexplained CO_2 change would likely be due to a factor that has minimal impacts on ocean O_2 and $\delta^{13}\text{C}$. One such factor would be whole-ocean alkalinity, which was likely higher at the LGM (Cartapanis et al., 2018; Omta et al., 2018; Weinans et al., 2021). However, the magnitude of this effect is difficult to quantify since we lack direct paleoproxies for alkalinity. We have included a brief paragraph about this in the Discussion section of the revised manuscript.

Lorraine Lisiecki

Reviewer #1 (Remarks on code availability):

The code runs and produces results that match what is shown in Figure 3.

Thank you for taking the time and effort to run and check our code.

References:

- Cartapanis, O., E.D. Galbraith, D. Bianchi, and S.L. Jaccard (2018), Climate of the Past 14: 1819–1850*
Omta, A.W., R. Ferrari and D. McGee (2018), Global Biogeochemical Cycles 32: 720–735
Weinans, E., A.W. Omta, G.A.K. van Voorn and E.H. van Nes (2021), Climate Dynamics 57: 523–535

Reviewer #2 (Remarks to the Author):

I was reviewer #2 for the initial submission. I think the authors have done a good job of responding to my comments, and have no further follow-up on these. I only have a few small remaining suggestions regarding the title and abstract.

We would like to thank Reviewer #2 for their thoughtful comments in both the previous and current Review rounds, which have helped improve the manuscript considerably. Below, we address the newest comments point-by-point.

I think the new title is too general – the authors only use stable carbon isotopes, and ‘constrain mechanisms’ is vague and leaves out the main point. I would suggest something like, “Global carbon isotope assessment indicates biological disequilibrium dominated ocean carbon storage at the LGM”.

That is a great point! We have decided on the title “Carbon isotope budget indicates biological disequilibrium dominated ocean carbon storage at the LGM” as a specific and to-the-point summary of our main message.

Abstract

I don’t feel this is accurate: ‘Here we show that stable carbon isotopes can be used to identify the driving mechanism.’ The isotopes do not allow identification of a single driving mechanism, they only help to characterize the carbon storage. Furthermore, this makes it sound like nobody used stable carbon isotopes for this problem before, when in fact they’ve been used for many decades. The novelty lies in the details of the calculation and size of the dataset. Please rephrase this.

We have removed this sentence, also to comply with Nature Communications’ 150 words limit for the Abstract.

I also think the abstract should mention the word disequilibrium somewhere. Perhaps something like: An analysis of the carbon isotopic signatures of different water masses indicates similar regenerated carbon inventories at the LGM and during the Holocene, **REQUIRING THAT THE CHANGE IN STORAGE WAS DOMINATED BY DISEQUILIBRIUM.**

We have incorporated this suggested textual modification.

Reviewer #3 (Remarks to the Author):

The authors have greatly improved their manuscript, in my opinion, based on all the reviewers' input, and I have no further comments, apart from one.

We would like to thank Reviewer #3 for their thoughtful comments in both the previous and current Review rounds, which have helped improve the manuscript considerably.

It seems the authors will not budge on this one, so I don't wish to make a big deal of it; however, I still do not agree that carbon retained by the ocean due to limited air-sea gas exchange can be referred to as 'biologically sequestered carbon'. I think it is clear that biology did not do the 'sequestering' in this case (even if it did the 'pumping'). If air-sea exchange was made more efficient (i.e. there was no sea ice, or the ocean circulation was altered etc.), but there was NO change in biological activity, the sequestration could be diminished or removed (i.e. abiotically).

The Reviewer is not disputing the picture we propose but rather our terminology. We agree that biology pumps carbon into the deep ocean and disequilibrium (due to sea ice, ocean circulation) helps to keep it there. In the revised manuscript, we have emphasized the key role of disequilibrium in the Abstract and in the new title: "Carbon isotope budget indicates biological disequilibrium dominated ocean carbon storage at the LGM" (inspired by Reviewer #2's suggestion).

That said, the point about avoiding conflation with solubility effects is a valid one, and I agree that my suggestions were therefore not perfect. I just feel that the chosen terminology is unfortunate in directing attention away from the physical processes that are actually doing the 'sequestering' and that are receiving renewed attention in this study – these are physical processes that will need to be studied further and therefore might benefit from being highlighted.

Since we could not find a better terminology, we decided to stick with "biologically sequestered carbon". By using the term "biologically sequestered" we wish to emphasize the isotopically light signature of the sequestered deep carbon, without implying that the sequestration was solely due to biology. To clarify this, we have included the following two sentences at the end of Section 2.1: "We are using the term biologically sequestered carbon, because the disequilibrium carbon has a primarily biological origin. This does not imply that the sequestration is driven solely by biology, as sea ice and the ocean circulation play key roles in maintaining the carbon disequilibrium."

In any event, I look forward to seeing the study in print, and discussing it further in other forums!

We are very much looking forward to such further discussions as well!